# Most mitochondrial dGTP is tightly bound to respiratory complex I through the NDUFA10 subunit

David Molina-Granada[1,2], Emiliano González-Vioque[1,2,5], Marris G. Dibley[3], Raquel Cabrera-Pérez[1,2], Antoni Vallbona-Garcia[1], Javier Torres-Torronteras[1,2], Leonid A. Sazanov [4], Michael T. Ryan[3], Yolanda Cámara [1,2,6✉] & Ramon Martí [1,2,6✉]

Imbalanced mitochondrial dNTP pools are known players in the pathogenesis of multiple human diseases. Here we show that, even under physiological conditions, dGTP is largely over-represented among other dNTPs in mitochondria of mouse tissues and human cultured cells. In addition, a vast majority of mitochondrial dGTP is tightly bound to NDUFA10, an accessory subunit of complex I of the mitochondrial respiratory chain. NDUFA10 shares a deoxyribonucleoside kinase (dNK) domain with deoxyribonucleoside kinases in the nucleotide salvage pathway, though no specific function beyond stabilizing the complex I holoenzyme has been described for this subunit. We mutated the dNK domain of NDUFA10 in human HEK-293T cells while preserving complex I assembly and activity. The NDUFA10[E160A/R161A] shows reduced dGTP binding capacity in vitro and leads to a 50% reduction in mitochondrial dGTP content, proving that most dGTP is directly bound to the dNK domain of NDUFA10. This interaction may represent a hitherto unknown mechanism regulating mitochondrial dNTP availability and linking oxidative metabolism to DNA maintenance.

[1] Research Group on Neuromuscular and Mitochondrial Disorders, Vall d'Hebron Institut de Recerca, Universitat Autònoma de Barcelona, Barcelona, Catalonia, Spain. [2] Biomedical Network Research Centre on Rare Diseases (CIBERER), Instituto de Salud Carlos III, Madrid, Spain. [3] Department of Biochemistry and Molecular Biology, Monash Biomedicine Discovery Institute, Monash University, Melbourne, Australia. [4] Institute of Science and Technology Austria, Klosterneuburg, Austria. [5] Present address: Department of Clinical Biochemistry, Hospital Universitario Puerta del Hierro-Majadahonda, Madrid, Spain. [6] These authors contributed equally: Yolanda Cámara, Ramon Martí. ✉email: yolanda.camara@vhir.org; ramon.marti@vhir.org

Mitochondria have their own genome, which encodes for key subunits of the respiratory chain complexes and the RNAs required for their expression. However, synthesis of all other oxidative phosphorylation system (OXPHOS) subunits and elements regulating mitochondrial DNA (mtDNA) expression and maintenance fully depends on nuclear-encoded proteins. Therefore, adaptation of mitochondrial function to physiological demands requires a highly regulated and efficient crosstalk between both genomes. It is essential for mitochondria to keep a balanced mitochondrial deoxyribonucleoside triphosphate (dNTP) pool in order to provide substrates to polymerases and other enzymes that act in the replication and reparation processes[1]. Mammalian cells obtain dNTPs from two different metabolic sources, cytosolic *de novo* synthesis, and salvage pathways. *De novo* synthesis depends on the activity of ribonucleotide reductase (RNR), which converts ribonucleoside diphosphates (rNDPs) to deoxyribonucleoside diphosphates (dNDPs), the latter being subsequently phosphorylated to dNTPs by nucleoside diphosphate kinases. However, this pathway is drastically reduced in post-mitotic tissues and, therefore, mtDNA replication and repair depend on the deoxyribonucleoside salvage pathway. This pathway consists of two parallel sets of enzymes located in the cytosol and mitochondrial matrix that catalyze the sequential phosphorylation of deoxyribonucleosides (dN), derived from diet or DNA turnover, to dNTPs[2].

Defects in genes encoding mitochondrial deoxyribonucleoside kinases and other dNTP-metabolism related enzymes (*RRM2B*, *TYMP*, *DGUOK*, and *TK2*) are associated with mtDNA depletion and multiple deletions syndromes in humans[3].

Accuracy of DNA synthesis depends on dNTP size but also on its relative composition. Nucleotide selectivity and proofreading, and thus DNA replication fidelity, are largely controlled by the relative concentrations of the four canonical dNTPs at the replication fork. Hence, anabolic and catabolic processes tightly regulate intracellular concentrations of these dNTPs, and imbalances in the pools can be genotoxic[4–7].

In the present study, we have found that dNTP pools from mouse tissue mitochondria are highly asymmetric even in physiological conditions, with dGTP being largely overrepresented with respect to the other three canonical dNTP species. The same dGTP asymmetry has been found in human cultured cells. A similar unevenness had been described earlier in rat tissue mitochondria[8]. However, it remained controversial whether overrepresentation of dGTP could be due to technical artefacts during dNTP extraction and determination, given that other laboratories reported much lower dGTP values with minimal asymmetries of the pool[9–11]. In our study, we have found that 90% of mitochondrial dGTP is bound to NDUFA10, an accessory subunit of complex I of the mitochondrial respiratory chain. Mitochondrial complex I is a boot-shaped enzyme formed by a hydrophilic arm that penetrates into the matrix, and a hydrophobic arm that is embedded in the inner mitochondrial membrane. The matrix arm is involved in NADH oxidation with electrons being transferred through iron sulfur clusters to ubiquinone, while the membrane arm undergoes conformational changes to allow translocation of protons to the intermembrane space, contributing to the electrochemical gradient used for the generation of ATP. Mutations in complex I subunits can lead to various defects such as reactive oxygen species (ROS) production, neurodegenerative diseases, apoptosis, and cell death[12–14]. Complex I is formed by 45 subunits in mammals, of which only 14 are considered core subunits essential for the catalytic function of the enzyme and already present in bacteria (7 of them mtDNA-encoded). Accessory or supernumerary subunits are in most cases required for full assembly of the complex[15]. Together with numerous required assembly factors, accessory subunits have evolved from pre-existing protein families. Some preserve conserved protein domains and functional motifs which suggest specific roles beyond redox-linked proton translocation[16,17].

The interaction of respiratory chain complex I with dGTP we report here may represent a hitherto unknown mechanism linking energy demands to dNTP availability.

## Results

**Mitochondrial dNTP pools are highly asymmetric in mouse tissues or human cells**. A balanced dNTP pool is considered necessary for maintaining DNA replication rate and fidelity[4–7]. However, we and others have previously described important asymmetries in the composition of the mitochondrial dNTP pool in physiological conditions[8,18,19]. In this study, we quantified dNTP pools in mitochondria from fresh brain and liver of adult C57Bl/6 J mice. After extraction with trichloroacetic acid (TCA) of mitochondrial dNTPs, we determined that dGTP was highly overrepresented in comparison to other dNTP species, being 67% and 73% of total mitochondrial dNTPs in liver and brain, respectively (Fig. 1a, b). We made a similar observation when analyzing mitochondrial dNTP pools in proliferating HEK-293T cells, where dGTP was again the most abundant dNTP (59% of total dNTPs) (Fig. 1c). We also quantified dNTP pools in whole HEK-293T cell extracts. Overall, dNTP levels were higher in whole cell than in mitochondrial extracts as referred to mg of protein, especially dTTP, in agreement with the proliferative state of HEK-293T cells that maintain elevated dNTP levels to provide sufficient substrate for DNA replication[5,20]. dTTP was 59% of total dNTPs, whereas much lower and comparable levels of dATP, dGTP, and dCTP were quantified (Fig. 1c). Our results suggest that although mitochondria and cytosol can exchange deoxyribonucleotides (dNMP, dNDP, or dNTP) across mitochondrial membrane[10,21,22], the exact composition of the pool may differ between both compartments even under proliferating conditions.

**Most mitochondrial dGTP is bound to protein**. Prior studies had also reported pronounced asymmetries in the mitochondrial dNTP pool from mammalian tissues, with a predominant presence of dGTP in most cases[8]. However, other groups had described extremely low dGTP levels in similar extracts[9,10]. We detected differences during the preparation of dNTP extracts that could explain these conflicting results. Classic nucleotide extraction procedures are based on an acid protein precipitation (typically TCA or perchloric acid, PCA), followed by neutralization, for instance, with a tri-octyl-amine solution in Freon[23,24]. Later on, the use of alternative methanol-protein precipitation was generalized over acid-precipitation in an attempt to prevent acid hydrolysis of nucleotides during extraction[25]. In this procedure, methanol incubation is followed by centrifugation to pellet down precipitated proteins. The procedure includes a short boiling step following methanol incubation aimed at the complete denaturation of methanol-resistant enzyme activities that could alter dNTP composition of the final extract[26]. While some authors ensure complete denaturation of all proteins in the sample by boiling prior to centrifugation (MeOH + B + C), others boil the supernatant obtained after centrifugation and thus devoid of most proteins (MeOH + C + B). However, regardless of the method of choice, some enzymes may still be active during sample processing and contribute at least partially to final dNTP quantification. Consistent with this idea, dNTP recovery after addition of a known amount of dNMPs, dNDPs, or dNTPs to mitochondrial lysates largely varied from the expected concentration, especially when following methanol-based extraction (Supplementary Fig. 1). This suggests that some kinases,

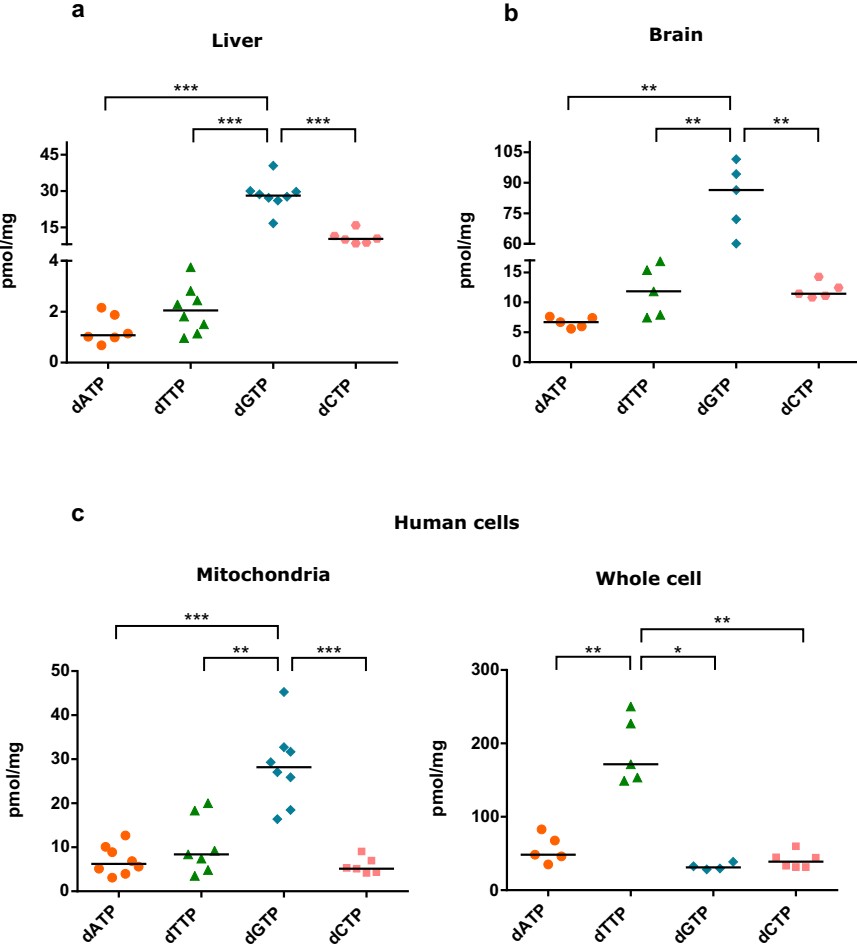

**Fig. 1 Mitochondrial dNTP pool is highly asymmetric. a** dNTP content in TCA extracts from mouse liver mitochondria (pmol dNTP/mg of protein). $N = 6$–8 independent experiments. **b** dNTP content in TCA extracts from mouse brain mitochondria (pmol dNTP/mg of protein). $N = 5$ independent experiments. **c** dNTP content in TCA extracts from HEK-293T mitochondria (left panel) and whole-cells (right panel) (pmol dNTP/mg of protein). $N = 7$–8 of independent experiments. Scatter plots represent the median (horizontal line). Two-tailed Mann-Whitney U test, $*p < 0.05$, $**p < 0.01$, $***p < 0.005$.

phosphatases, and presumably other enzymes are still active during extraction and may alter the final dNTP concentration. Amongst all tested methods, acid-based extractions yielded the best recovery ratios for all dNTPs with negligible acid hydrolysis of added dNTP (Supplementary Fig. 1).

We prepared extracts following the various methods and quantified dNTP species in fresh mouse liver mitochondria. Different extraction methods could account for small variations in dNTP stability or solubility. Nonetheless, we obtained similar results using either TCA, PCA acid-extraction, or MeOH + B + C, albeit with slightly lower levels of all dNTPs when using the latter approach (Fig. 2a). We confirmed preferential abundance of dGTP (Fig. 1a) in extracts prepared following either of the three methods. However, while both methanol-based methods yielded comparable levels of dATP, dTTP, and dCTP, we barely detected dGTP (levels below our quantification limit) when using MeOH + C + B, suggesting almost all dGTP in the sample was lost during centrifugation of methanol extracts that had not been previously boiled. Conversely, boiling of methanol extracts before centrifugation (MeOH + B + C) released dGTP from the protein fraction to levels comparable to those observed in acid-extracts. (Fig. 2a). To confirm that most dGTP was lost with the precipitate in MeOH + C + B extracts, we used a modified method in which we used TCA to extract dNTPs from both the pellet and supernatant fractions following methanol incubation and centrifugation. Both in liver and

brain, protein-bound/free dNTP ratios (pellet/supernatant) indicated that dATP, dTTP, and dCTP were mostly free (ratio < 1) in mitochondria (Fig. 2b). However, ratios for dGTP were always well above 1, showing that a vast majority of the nucleotide was bound to protein in mitochondria from both tissues (97% in liver and 87% in brain). These results indicate that most dGTP is likely bound to protein in tissue mitochondria. Furthermore, dGTP release from the protein-precipitated fraction occurred only after severe denaturing treatment (boiling or acid), suggesting the nucleotide-protein interaction was tight and stable throughout mitochondria isolation and methanol treatment.

We obtained further evidence of dGTP interaction with mitochondrial proteins by additional radiolabeling approaches. Electrophoretic mobility of [α-$^{32}$P]-dGTP was retarded after incubation with protein lysates from fresh mouse liver mitochondria (Fig. 2c) as detected by EMSA. We detected two shifted bands that were competed away by an excess of cold dGTP (black arrowheads). These bands were partially competed by a 100-fold molar excess of different guanine-containing nucleotides (GTP, dGMP, and especially dGDP) but not by a 100-fold molar excess of any of the other dNTPs (dATP, dCTP, or dTTP), indicating that they likely corresponded to dGTP-protein complexes in the mitochondrial lysate.

We performed a similar experiment, but this time resolving the nucleotide-protein complexes in BN-PAGE gels (Figs. 2d, e).

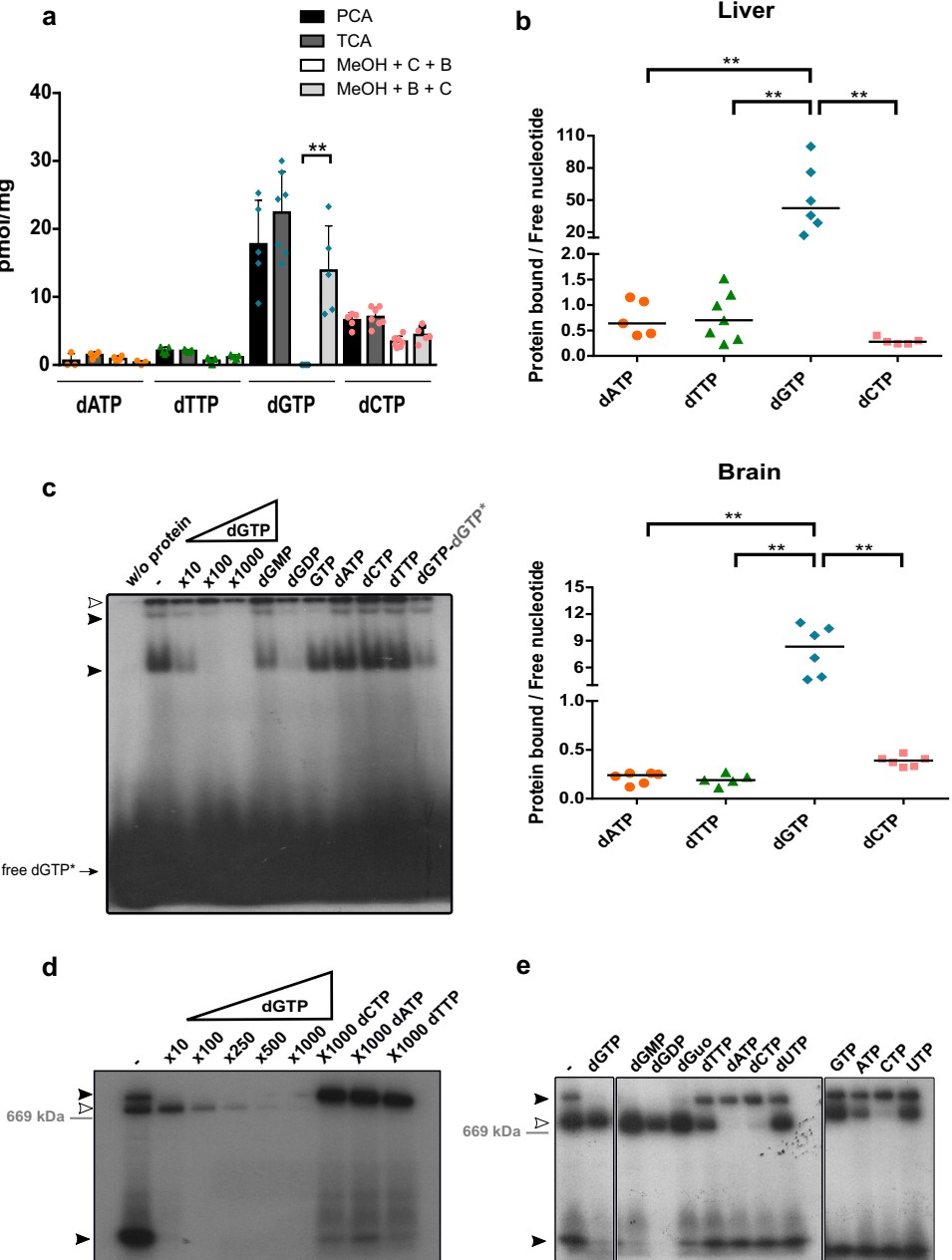

Again, we identified radiolabeled complexes that were specifically competed by an excess of cold dGTP or deoxyguanosine-containing nucleotides and deoxyguanosine (black arrowheads). As with EMSA, the retarded bands were resistant to an excess of other dNTPs and also ribonucleotides (including GTP) (Fig. 2e). One of the identified bands ran above 669 kDa, thus corresponding to a high-molecular-weight dGTP-protein complex.

**dGTP is bound to respiratory chain complex I through NDUFA10 subunit.** In order to identify which protein(s) were complexed with mitochondrial dGTP, we performed affinity chromatography with immobilized ɣ-amino-octyl-dGTP on mouse liver mitochondrial lysates prepared under native conditions (1% n-docdecyl ß-D-maltoside, DDM). Chromatography with immobilized ɣ-amino-octyl-dCTP and blank resin was run in parallel for the same lysates as a control for unspecific interactions with the ɣ-amino-octyl linker and agarose bed. Pulled-down proteins were later resolved by SDS-PAGE gels and visualized by silver-staining. We detected two main bands specifically purified with ɣ-amino-octyl-dGTP at molecular weights around 37 and 25 kDa (Fig. 3a). We identified proteins in these bands in a second large-scale pull-down with only ɣ-amino-octyl-dGTP and blank agarose coupled to LC-ESI-MS/MS as the complex I subunit NADH:ubiquinone oxidoreductase A10 (NDUFA10) (UniProtKB - Q99LC3) and deoxyguanosine kinase (dGK) (UniProtKB - Q9QX60) (Fig. 3a). We also analyzed two minority bands at around 75 and 15 kDa primarily containing a known GTP-binding protein and other subunits of complex I (Supplementary Fig. 2 and Supplementary Table 1). In fact, numerous other subunits of complex I were present in the different bands, although at a much lower relative abundance compared to that of NDUFA10. This enrichment was possibly due to the lysate being obtained under native conditions that would favor NDUFA10 co-purification with other complex I constituting proteins (Supplementary Fig. 2 and Supplementary Table 1).

**Fig. 2 A substantial fraction of mitochondrial dGTP is bound to protein. a** Comparison between different dNTP extraction methods. dNTP levels (pmol dNTP/mg of protein) in liver mitochondria extracts prepared with acid (0.6 M PCA or 0.5 M TCA), 60% methanol treatment, followed by centrifugation and then boiling the supernatant (MeOH + C + B), or 60% methanol treatment followed by boiling before centrifugation (MeOH + B + C). dGTP values obtained after MeOH + C + B extraction were significantly different from values obtained by any other tested method. Results are mean values of N independent experiments (symbols): PCA = 3–5; TCA = 4–7; MeOH + C + B = 4–7; MeOH + B + C = 3–5. Error bars represent standard deviation. **b** Ratio of protein bound to free dNTP fractions in mouse liver (top plot) and brain (bottom plot) mitochondria. Fractions were obtained after treatment of mitochondria with 60% methanol, centrifugation, and complete denaturation of resulting pellets (protein-bound fraction) and supernatants (free fraction) with TCA. $N = 5$–7 (top plot) and 5–6 (bottom plot) of independent experiments **c** Electrophoretic mobility shift assay (EMSA) of radiolabeled [$\alpha$-$^{32}$P]-dGTP (0.06 pmol) after incubation with 10 µg of native protein lysates from fresh mouse liver mitochondria. Specificity of shifted bands was tested against a 10- to 1,000-fold molar excess of cold dGTP, a 100-fold molar excess of the other cold dNTPs (dATP, dTTP, and dCTP) or different guanine-containing nucleotides (dGDP, dGMP, and GTP). Competitors were incubated with the protein lysate for 30 min prior to the addition of [$\alpha$-$^{32}$P]-dGTP (last lane resolves an assay where radiolabeled dGTP was incubated before cold dGTP addition). Black arrows indicate specific dGTP-protein complexes. The white arrow tags unspecific [$\alpha$-$^{32}$P]-dGTP binding. **d** Autoradiography of [$\alpha$-$^{32}$P]-dGTP-protein complexes resolved by BN-PAGE. 30 µg of native protein lysates from fresh mouse liver mitochondria were incubated with 1 µM [$\alpha$-$^{32}$P]-dGTP (300 Ci/mmol) and resolved by blue native-polyacrylamide gel electrophoresis (BN-PAGE). Competition assays were performed by pre-incubating the lysates with an excess of 10- to 1,000-fold of cold dGTP or 1,000-fold of dATP, dTTP, and dCTP. A white arrow indicates a signal from dNTP-protein complexes (not dGTP-specific). Black arrows indicate specific dGTP-protein complexes as revealed by subsequent autoradiography of dried BN-PAGE gels. Migration of a 669 kDa MW marker is indicated. Image shows representative data of at least 2 independent experiments **e** Analysis of retardation in electrophoretic mobility of radiolabeled [$\alpha$-$^{32}$P]-dGTP (300 Ci/mmol) after incubation with mouse liver mitochondria lysed under native conditions and resolved by BN-PAGE gel. Several competition assays were performed by pre-incubating the lysate with a 100-fold molar excess of different guanine-containing nucleotides (dGTP, dGDP, dGMP, and GTP), dNTPs, rNTPs, and dGuo. Lanes 1–9 were discontinuously loaded in one same gel while lanes 10–13 were loaded in a second gel. Both gels were run simultaneously with the same sample treated with the different competitors. Two-tailed Mann-Whitney U test, *$p < 0.05$, **$p < 0.01$, ***$p < 0.005$.

dGK is a nucleoside salvage kinase that catalyzes the phosphorylation of purine deoxyribonucleosides within mitochondria. NDUFA10 is an accessory subunit sitting at the junction between the matrix and the membrane arm of mitochondrial respiratory complex I. While NDUFA10 has no direct enzymatic function, it is critical for assembly[15] and may be important for stability of the active state of complex I[27]. Whereas dGTP is a known modulator of dGK activity[28], no interaction of dGTP with NDUFA10, or any other subunit of complex I, had been earlier described. However, NDUFA10 contains a dNK domain shared with deoxyribonucleoside kinases (Supplementary Fig. 3), such as deoxycytidine kinase, thymidine kinase 2, or dGK itself[29–31]. In fact, recent cryo-electron microscopy (cryo-EM) data on mouse NDUFA10 structure has suggested binding of a two-phosphate purine nucleotide to the dNK domain of NDUFA10 in mouse complex I[27]. Agip et al. proposed ADP as the purine nucleotide binding NDUFA10, based on its considerable abundance in the mitochondrial matrix (Fig. 4c). On the other hand, in ovine complex I[32] an AMP molecule has been modelled in this area, along with phosphorylated NDUFA10 S36 residue in the vicinity (Fig. 4a), on the basis of high-resolution cryo-EM density and mass-spectrometry data for bovine enzyme[27]. We performed a second pull-down with γ-amino-octyl-dGTP on mouse liver mitochondrial native extracts and monitored NDUFA10 binding using different purine ribonucleotides, including ADP, as eluents. We found that only guanine deoxyribonucleotides (dGTP, dGDP, and dGMP) and dGuo could effectively elute NDUFA10 from immobilized dGTP (Fig. 3b). Elution efficacy decreased proportionately with lower phosphorylation states. Re-examination of the cryo-EM density both for ovine and mouse complex I suggests that in principle, dGTP can also fit in this area. For ovine enzyme the fitting would be optimal if S36 would be non-phosphorylated (Fig. 4b). S36 phosphorylation has only been shown for bovine complex I. There is no mass-spectroscopy data on the phosphorylation state of S36 in ovine complexes, and this residue is in fact, substituted by asparagine or glycine in the mouse, porcine and human homologs. Within the mouse structure, dGTP would fit with its gamma phosphate facing sideways (Fig. 4d) while in the ovine structure, this

phosphate may replace the one putatively attached to S36 (Fig. 4b). In addition, dGTP could still fit within the ovine structure with phosphorylated-S36, in a similar mode to that suggested for mouse complex I (not shown). Therefore, the available structural data is in principle consistent with the possibility of dGTP binding to NDUFA10 subunit.

We further assessed whether endogenous dGTP was interacting with complex I in mitochondria. First, we analyzed dGTP-containing complexes from DDM-solubilized mitochondrial lysates using sucrose density gradient centrifugation. We monitored all fractions for migration of the various respiratory chain complexes and endogenous dNTP levels (Fig. 3c). Our results showed that most mitochondrial dGTP (70%) co-sedimented with respiratory chain complex I in both brain (fractions 2–3) and liver (fractions 2–4), while the other dNTPs sedimented at lower density fractions (dATP was below our lower detection limit at all fractions, presumably due to its degradation during lysate preparation). We also located mitochondrial transcription factor A (TFAM), usually associated with mtDNA[33], at low density fractions. Likewise, precise co-sedimentation of dGTP with NDUFA10 along the density gradient was maintained when using a stronger non-ionic detergent such as 1% Triton-X-100 for lysate preparation (Supplementary Fig. 4).

To confirm dGTP interaction with complex I, we immuno-precipitated the whole complex from mouse liver mitochondrial lysates prepared in native conditions and determined the presence of associated dNTPs in TCA-extracted samples. Immunoprecipitation (IP) fractions were monitored by SDS-PAGE and immunodetection with several antibodies against different respiratory chain subunits (Fig. 3d). Approximately 70% of all dGTP present in the input lysate co-immunoprecipitated with complex I. On the other hand, dGTP was barely detected in the flow-through fractions virtually depleted of complex I. Values for the other three dNTPs in the input lysate were undetectable, possibly due to degradation during the experimental procedure (Fig. 3d).

Our previous radiolabeling assays suggested that dGTP was, at least in part, interacting with a high-molecular-weight complex compatible with respiratory complex I in mitochondrial lysates (Fig. 2d, e). Immunodepleted lysates (flow-through after IP)

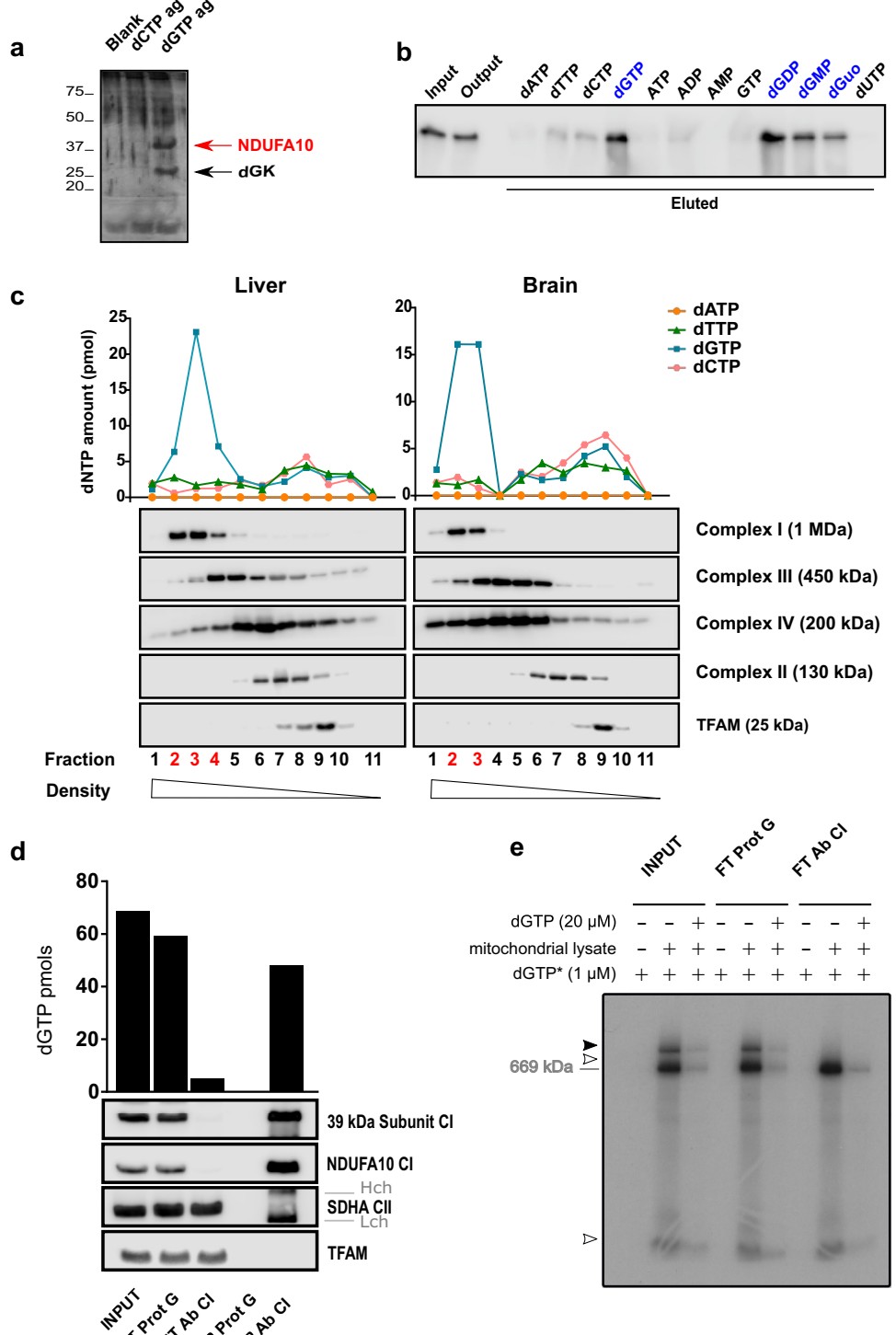

allowed us to recognize this high-molecular-weight complex as the one formed by complex I and radiolabeled dGTP (black arrowhead; Fig. 3e). Thus, although the protein-nucleotide interaction was tight and stable through preparation of mitochondrial lysates, it was dynamic and allowed certain degree of exchange with exogenous dGTP, as observed in our pull-down and radiolabeling experiments.

At this point, our data indicated that most mitochondrial dGTP is tightly bound to complex I, anchored to the inner mitochondrial membrane rather than free in the organelle matrix. To confirm that dGTP binding to complex I was via the NDUFA10 subunit, we performed photolabeling assays with purified protein and radioactive dNTPs.

For this purpose, we transformed HeLa cells to stably express recombinant human NDUFA10 fused to a Flag-tag in its C-terminus (NDUFA10[FLAG]). We purified NDUFA10[FLAG] by IP with M2-antiFLAG antibody (Fig. 5a) and UV-crosslinked it to all four [α-$^{32}$P]-dNTPs (Fig. 5b). Photoaffinity labeling demonstrated NDUFA10 ability to bind dGTP in vitro. This interaction was specific for dGTP, given that no binding was detected for either dATP, dTTP, or dCTP (Fig. 5b). A further UV-crosslink assay was performed with immunopurified NDUFA10[FLAG], [α-

**Fig. 3 Most mitochondrial dGTP is bound to respiratory chain complex I through NDUFA10 subunit. a** Silver staining of a representative affinity pull-down with a native mouse liver mitochondrial lysate on agarose-immobilized dGTP (γ-amino-octyl-dGTP; dGTPag), dCTP (γ-amino-octyl-dCTP; dCTPag), and blank resin (blank). Pulled-down proteins were eluted from the resins with 100 mM dGTP. NDUFA10 and dGK proteins were identified on subsequent analysis by LC-MS/MS of the two majority bands pulled-down with dGTP (arrows). Image shows representative data of 2 independent experiments. **b** Western blot analysis of NDUFA10 pulled-down with γ-amino-octyl-dGTP on native mouse liver mitochondrial extracts. NDUFA10 was eluted from the resin with 100 mM of different dNTPs and purine ribonucleotides. **c** Analysis of dNTP and OXPHOS complexes migration on sucrose density gradients. Native lysates (from 5 mg of protein) of mouse tissue mitochondria in liver (left panel) and brain (right panel) were resolved in 15–37.5% sucrose density gradients. Individual fractions were subsequently processed for both dNTP and protein determination. The top graph shows dNTP levels (pmol) in the different fractions. Image panels show representative subunits of the different OXPHOS complexes (NDUFA10 [complex I], UQCRC2 [complex III], Cox IV [complex IV], SDHA [complex II]) and TFAM in the different fractions as analyzed by western blot and subsequent immunodetection with specific antibodies (image panels). Fractions containing both dGTP and complex I are highlighted in red. Shown data is representative of 2 independent experiments **d** Analysis of dGTP co-IP with native complex I. Five milligrams of protein of mouse liver mitochondria were immunoprecipitated with a specific antibody recognizing the full complex, previously conjugated to agarose beads (Ab CI). Equivalent protein G (prot G) beads were used as a control for unspecific agarose-binding. The top graph shows dGTP amount (pmol) in each input, flow-through (FT) and eluate (IP) fraction, after TCA extraction. Bottom image panels show the respective western blot analyses: anti-39 kDa subunit (complex I), anti-NDUFA10 (complex I), anti-SDHA (complex II), and anti-TFAM antibodies. Cross-reactivity with monoclonal antibody heavy (Hch) and light (Lch) chains did not interfere with the proteins being analyzed. Shown data is representative of 2 independent experiments **e** Autoradiography of [α-$^{32}$P]-dGTP-protein complexes on the input and complex I immunodepleted lysates resolved by BN-PAGE (input and flow-throughs after complex I and control protein G immunoprecipitations used in Fig. 3d). Specificity of detected bands was tested against a 20-fold molar excess of cold dGTP. The black arrow indicates dGTP specific binding to complex I. White arrows indicate dGTP-protein interactions other than with full complex I or unspecific binding of radiolabeled nucleotide (not dGTP competed). Migration of a 669 kDa MW marker is indicated.

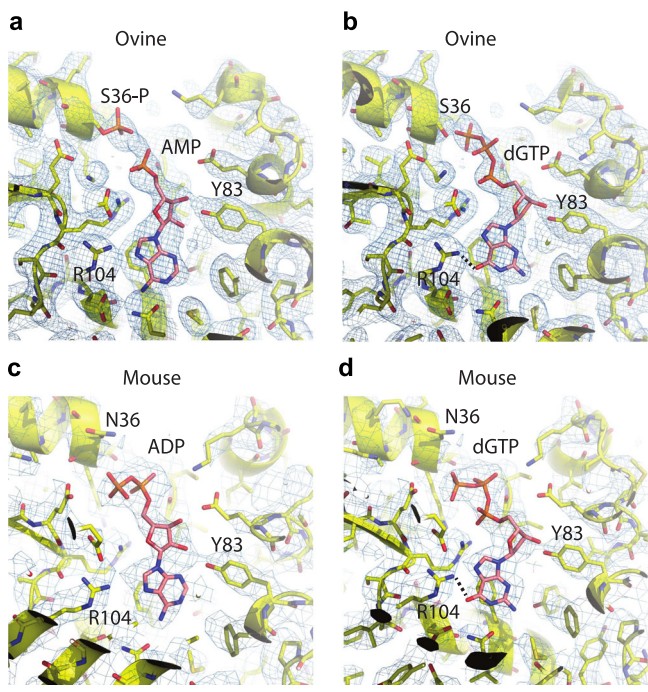

**Fig. 4 Putative mode of dGTP binding to mammalian complex I. a, c** Cryo-EM maps and models for ovine (PDB 6ZKA, 2.5 Å resolution) and mouse (PDB 6g2j, 3.3 Å resolution) complex I. Cryo-EM density is shown as a blue mesh. **b, d** Putative models of dGTP fit into the same density. Key residues discussed in the text are labelled.

$^{32}$P]-dGTP, and a set of cold competing nucleotides (Fig. 5c). Data showed only deoxyguanosine-containing competitors could partially compete for dGTP binding to NDUFA10. These data were in agreement with previous binding assays performed under native conditions for whole complex I (Figs. 2c, e, and 3b) and proved NDUFA10 interaction with dGTP to be highly selective.

Finally, we performed an approximation to the binding stoichiometry of the complex NDUFA10-dGTP. First, we ran an SDS-PAGE with an aliquot of immunopurified NDUFA10-FLAG protein along with a set of bovine serum albumin (BSA)

standards. Following Coomassie Brilliant Blue G250 staining, and densitometry of revealed bands, we extrapolated NDUFA10$^{FLAG}$ concentration from the BSA standard calibration curve (Fig. 5d). This now-quantified NDUFA10$^{FLAG}$ protein was used to generate a second calibration curve. We loaded known amounts of mouse liver mitochondrial lysate along with the NDUFA10$^{FLAG}$ standards and detected NDUFA10 by western blot. After densitometry of immunodetected bands, we extrapolated that 1.87 ± 0.35 µg (mean ± SD) of NDUFA10 were present in 1 mg of mitochondrial protein lysate (Fig. 5d). This corresponds to 44.5 pmol of NDUFA10 per mg of mitochondrial protein (NDUFA10 molecular weight = 42,000 g/mol). Based on our data, there are 28.2 ± 6.5 pmol of dGTP (mean ± SD) per mg of mitochondrial protein (Fig. 1a) and 98% of mitochondrial dGTP is bound to NDUFA10 in mouse liver mitochondria (Fig. 2b). Therefore, we could estimate that there is 1 pmol of dGTP per 1.6 pmol of NDUFA10 in liver mitochondria. Available data on NDUFA10 structure suggests a single nucleotide binding site in the protein[27], thus the most probable protein:nucleotide stoichiometry ratio in mouse liver mitochondria is approximately 1:1.

**dGTP binds to the dNK domain of NDUFA10.** Finally, we aimed to analyze the impact that NDUFA10-dGTP interaction had on mitochondrial dNTP pools. To do so, we used a previously generated HEK-293T *NDUFA10* knockout (KO) cell line (NDUFA10$^{KO}$)[15]. As with most accessory subunits of complex I, NDUFA10 is essential for the whole complex assembly[15]. Hence, mutations affecting NDUFA10 stability lead to Leigh syndrome and severe isolated complex I deficiency[34–37]. Furthermore, available data suggest severe respiratory deficiency may substantially alter dNTP homeostasis[38,39]. Thus, to specifically analyze the biochemical consequences of disrupted NDUFA10-dGTP interaction, we also generated a cellular model in which dGTP binding could be impaired while preserving complex I assembly.

As previously mentioned, NDUFA10 shows a high percentage of identity with the deoxyribonucleoside kinase family with whom it shares the dNK domain, as registered in the PFAM database (PF01712). This domain is common to nucleoside salvage pathway kinases, such as dGK, deoxycytidine kinase (dCK), and thymidine kinase 2 (TK2) (Supplementary Fig. 3). Most amino acids surrounding the nucleoside binding pocket and the phosphate donor site (P-loop) are conserved in NDUFA10. All critical

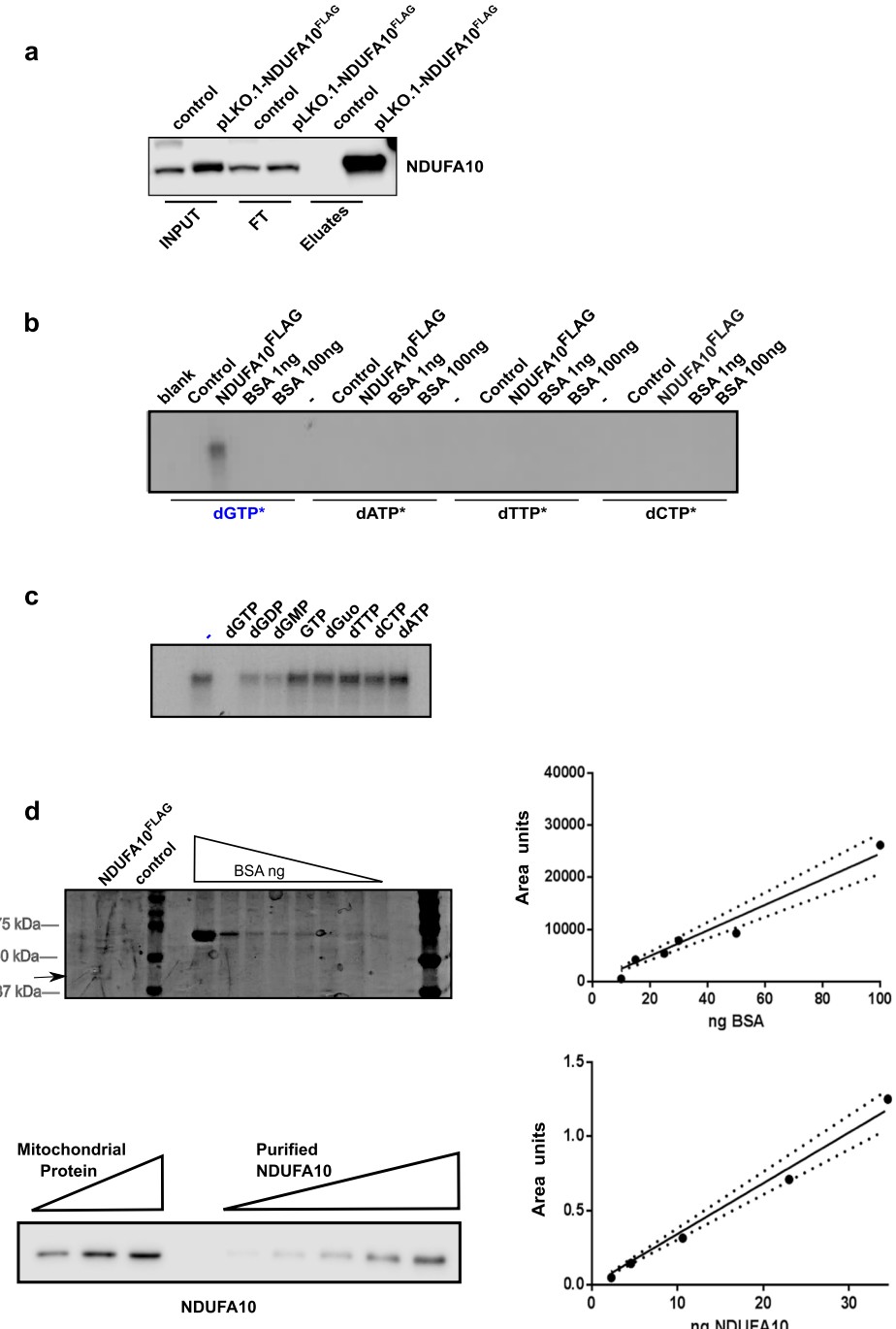

**Fig. 5 NDUFA10 nucleotide binding capacity is specific for dGTP. a** Western blot analysis of NDUFA10 IP with anti-FLAG M2 antibody on NDUFA10[FLAG] HeLa cell protein lysates. Lysates from HeLa cells transformed with pLKO.1-empty vector were used as a control. Image shows immunodetection with anti-NDUFA10 antibody on collected IP fractions (input, flow-through and eluates) were tested with anti-NDUFA10 antibody. **b** Autoradiography of a photoaffinity labeling assay with immunopurified NDUFA10[FLAG] and [α-32P]-dNTP. Eluates from HeLa cells transformed with empty pLKO.1 (control) and 1 ng or 100 ng of BSA were used to test for unspecific binding **c** Autoradiography image of a photoaffinity labeling assay with immunopurified NDUFA10[FLAG], [α-32P]-dGTP, and a 100-fold molar excess of cold dNTPs (dATP, dTTP, dGTP, and dCTP), different guanine-containing nucleotides (dGDP, dGMP, and GTP) and dGuo. **d** Relative quantification of NDUFA10 in mouse liver mitochondria. 15 μL of immunopurified NDUFA10[FLAG] (black arrow) were resolved next to a BSA standard curve (200, 100, 50, 30, 25, 20, 15, 10, and 0 ng) in a 12% SDS-PAGE. After Coomassie blue-G250 staining, the concentration of immunopurified NDUFA10[FLAG] was inferred from the BSA standard regression curve (area units/ng BSA) $y = 245.31x$; $R^2 = 0.9594$ (right top graph). A growing curve of immunopurified NDUFA10[FLAG] (2.3, 4.6, 10.6, 23, and 34.5 ng) was subsequently loaded next to a mouse mitochondrial liver lysate (10, 20, and 30 μg of protein) in a 12% SDS-PAGE (left bottom panel). NDUFA10 was detected by subsequent western blotting using anti-NDUFA10 specific antibody. The resulting regression curve (area units/ng NDUFA10[FLAG]) was used to determine the amount of NDUFA10 in liver mitochondrial lysate; $y = 0.034x$; $R^2 = 0.9815$ (right bottom graph). Dots represent mean values in the regression curve. The discontinuous lines depict the associated standard error.

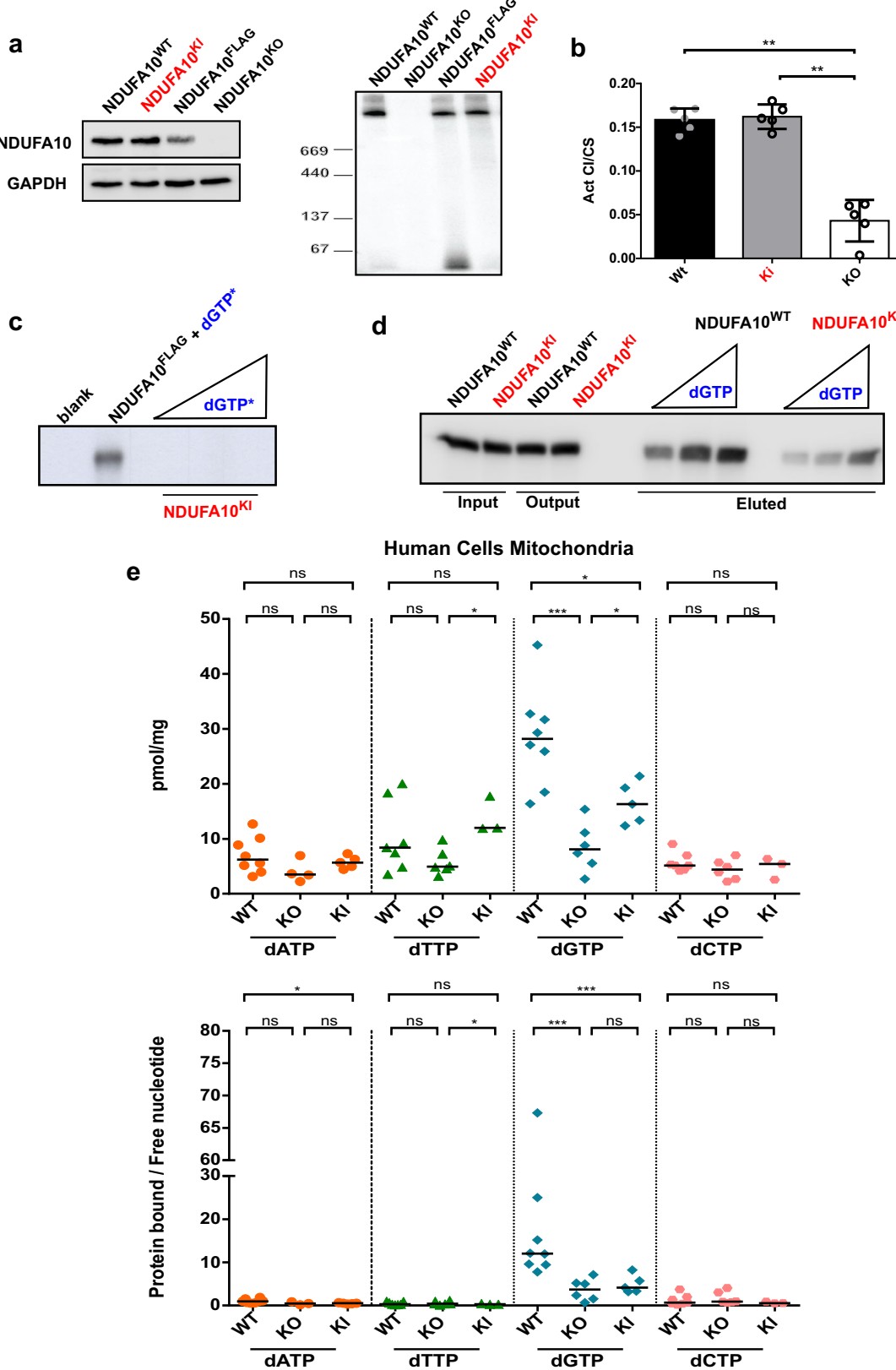

residues for nucleotide-interaction according to Cryo-EM data are present in mouse and human NDUFA10 (K70, E88, R139, E160, R161, K221, and E226)[27]. Thus, we introduced two missense mutations in the Glu-Arg-Ser (ERS) motif of human NDUFA10, E160A, and R161A, aimed at disrupting dGTP binding while preserving protein stability and thus complex I assembly. By

comparison with deoxycytidine kinase, E160 is predicted to directly interact with the magnesium ion required for catalysis and R161 would be part of the active site that is believed to play a role in the deprotonation of the 5'-OH group of the deoxyribose[40]. Mutations affecting the equivalent position in dGK showed decreased affinity for substrates and low catalytic activity[41,42].

**Fig. 6 dGTP binds to the dNK domain of NDUFA10 subunit. a** Analysis of NDUFA10 protein levels in whole cell lysates (by western blot; left panel) and in mitochondria (by BN-PAGE; right panel) of wild type (NDUFA10$^{WT}$), NDUFA10 knock-out (NDUFA10$^{KO}$), and NDUFA10 knock-in (NDUFA10$^{KI}$) HEK-293T cells, and NDUFA10$^{FLAG}$ HeLa cells. Immunodetection with anti-NDUFA10 and anti-GAPDH antibodies. **b** Ratio between complex I enzyme activity and citrate synthase activity in NDUFA10$^{WT}$, NDUFA10$^{KI}$, and NDUFA10$^{KO}$ cells. Results are mean values of 5 independent experiments (circles). Error bars represent standard deviations. **c** Autoradiography image of photoaffinity labeling with immunopurified NDUFA10$^{KI}$ and growing concentrations of [α-$^{32}$P]-dGTP (20 nM, 200 nM, and 2 μM). NDUFA10$^{FLAG}$ was incubated with 20 nM [α-$^{32}$P]-dGTP. **d** Western blot analysis of NDUFA10 pulled-down with γ-amino-octyl-dGTP on native mitochondrial extracts from NDUFA10$^{WT}$ and NDUFA10$^{FLAG}$ cells. Elution was performed using growing concentrations of dGTP (0.1, 1, and 10 mM) and immunodetection with anti-NDUFA10 antibody. **e** Mitochondrial dNTP levels in NDUFA10$^{WT}$, NDUFA10$^{KO}$, and NDUFA10$^{KI}$ cells. Total dNTPs in TCA extracts (pmol dNTP/mg of protein) (upper panel) and ratio of protein bound to free dNTP fractions (bottom panel). $N = 3$–8 independent experiments. Two-tailed Mann-Whitney U test, *$p < 0.05$, **$p < 0.01$, ***$p < 0.005$.

To generate a stable mutant cell line, the cDNA for flag-tagged NDUFA10$^{E160A/R161A}$ knock-in (KI) protein was introduced in NDUFA10$^{KO}$ cells by means of a lentiviral vector. We selected mutant clones with NDUFA10$^{E160A/R161A}$ expression levels close to those of endogenous NDUFA10 in wild-type (WT) HEK-293T cells (Fig. 6a). We verified that the inserted mutations affected neither assembly (Fig. 6a) nor activity of complex I (Fig. 6b). We also detected no altered oxygen consumption in agreement with preserved respiratory activity of the complex containing NDUFA10$^{E160A/R161A}$ (Supplementary Fig. 5)

We performed a photoaffinity labeling by UV-crosslink with [α-$^{32}$P]-dGTP and different cold competing nucleotides on flag-tagged immunopurified NDUFA10$^{E160A/R161AA}$ and wild-type protein (NDUFA$^{FLAG}$). In our in vitro conditions, we were unable to detect any dGTP binding for the mutant protein (Fig. 6c).

To confirm that the introduced mutations have a detrimental effect on NDUFA10 interaction with dGTP, we performed a new γ-amino-octyl-dGTP affinity pull-down with mitochondrial lysates from WT and KI cells using growing concentrations of dGTP as an eluent. Our results showed that the WT protein eluted more efficiently and at a lower dGTP concentration from the resin than the KI protein (Fig. 6d). These results suggested that NDUFA10$^{E160A/R161AA}$ was still able to interact with dGTP, though with a lower affinity than the WT protein. The discrepancy with previous UV-crosslinking results where no dGTP binding was detected for the purified mutant protein (Fig. 6c) is likely explained by native extracts in the pull-down providing richer and more favorable physicochemical conditions for the protein-dNTP interaction to occur.

Finally, we studied the consequences that the inserted mutations had on mitochondrial dNTP pools. First, we TCA-extracted dNTPs from mitochondria of WT, KO, and KI cells (Fig. 6e). We observed a marked decrease in mitochondrial dGTP levels for KI (50% reduction), and especially for KO cells (75% reduction). These results are consistent with most mitochondrial dGTP being associated to NDUFA10, and with the mutant NDUFA10$^{E160A/R161AA}$ protein retaining some capability to bind dGTP. Accordingly, we detected a decrease in the protein-bound to free nucleotide ratio (pellet/supernatant) for both KO and KI cells (Fig. 6e). As expected, we detected no significant changes for the other dNTPs in either KI or KO cells that were found mostly free, not associated to protein. The reduction of the dGTP pool did not induce mtDNA depletion or accumulation of multiple deletions in either KO or KI cells (Supplementary Fig. 6).

These results confirm that a vast majority of mitochondrial dGTP is tightly associated to the dNK domain of NDUFA10, and thus to mitochondrial complex I.

Based on available Cryo-EM data, dGTP binding to the dNK domain of NDUFA10 likely resembles a feedback inhibition mode, with the base and deoxyribose positioned as a deoxyribonucleoside substrate and the phosphate groups occupying the phosphate donor site[27,43]. Deoxyribonucleoside kinases are usually feedback-inhibited by end-products of their preferred

substrates[28], suggesting that NDUFA10 could have kinase activity with deoxyguanosine as main substrate. Therefore, we performed a second purification of NDUFA10$^{FLAG}$ from human HEK-293T cells to investigate potential kinase activity of NDUFA10. To help release bound dGTP, we incubated with a high concentration of deoxyguanosine (1 mM) prior to final IP. We performed an in vitro kinase assay with 15 ng of purified protein, ATP as phosphate donor, and 20 μM of radiolabeled deoxyguanosine as substrate. We could not detect any phosphorylation activity under our experimental conditions. However, we observed that our assay was limited by a partial inhibition from components in the IP elution buffer. Although further research is needed to fully rule out phosphorylation capacity of NDUFA10, this observation suggests an alternative role for the protein in mitochondrial physiology.

## Discussion

Composition of dNTP pools is tightly regulated in eukaryote cells. In fact, chemical and genetic disturbances affecting dNTP balance have long been known to damage genome integrity[4–7]. In mitochondria, genetic defects disturbing dNTP homeostasis (i.e., mutations in *TK2, DGUOK, TYMP, RRM2B*) interfere with mtDNA replication leading to qualitative and quantitative mtDNA defects[3]. However, we have found that the mitochondrial content of the four canonical dNTPs is largely asymmetrical, even under physiological conditions. In mitochondria from mouse brain and liver, the different dNTP species show variable levels, with dGTP being highly overrepresented. We have detected a similar bias in dNTP proportion in mitochondria from human replicative cells. Song et al[8]. already found that rat mitochondria had higher concentrations of dGTP than other dNTPs, and proposed a role for this asymmetry in the modulation of the physiological mutation rate in different tissues. However, these asymmetries were attributed to technical artefacts in nucleotide extract preparation or dNTP determination in related literature, and have remained controversial until today[9–11]. After comparing several methodologies for dNTP extraction, we have concluded that differences in the deproteinization protocols account for discrepancies in previously reported dGTP values. High values of dGTP were only obtained after harsh deproteinization methods involving sample denaturation (acid extraction or methanol plus high temperatures (boiling) prior to protein precipitation), which means an important part of the nucleotide was lost with the protein fraction in milder procedures. Here, we prove that a vast majority of dGTP is in fact stably and tightly bound to protein in mitochondria from healthy mouse tissues and cells. These findings will necessarily force a reinterpretation of previously available data on mitochondrial dNTP pools.

With further experiments, we demonstrated that exchange with the free nucleotide pool was still possible for the binding protein(s), allowing us to identify NDUFA10 as the main candidate to sequester mitochondrial dGTP by nucleotide-affinity purification. Our data showing dGTP co-immunoprecipitation (co-

IP) and co-sedimentation in density gradients with native complex I from mouse tissue proved that NDUFA10 interaction with the nucleotide is stable and occurs in physiological conditions. Importantly, this interaction involves most (70%) of the endogenous content in mitochondrial dGTP.

NDUFA10 is one of the five accessory subunits of complex I that are only present in metazoans. It is a 42 kDa subunit located in the matrix arm of complex I (ND2 domain), and its mutations are associated with an early-onset progressive neurodegenerative disorder (Leigh syndrome)[34–37]. No specific roles other than contributing to full assembly of the complex have been described for this protein[15], although recent data suggest it may stabilize the active state of the enzyme[27]. NDUFA10 is also one of the subunits that are phosphorylated in complex I[44]. Phosphorylation at S250 depends on PINK1[45], a serine/threonine-protein kinase involved in the activation of autophagy and Parkinson's disease. It is not clear how this modification is related to complex I deficiency in PINK1 mutant cells, but a possible influence on NDUFA10 stability and thus complex I assembly has been proposed[46]. Interestingly, enhancing nucleotide metabolism (by deoxyribonucleoside or folic acid supplementation) rescues mitochondrial dysfunction and neurodegeneration in a PINK1 model of Parkinson's disease[47]. A possible interdependency of NDUFA10 phosphorylation and dGTP binding capacity needs to be addressed in further research.

NDUFA10 shares a functional domain (domain dNK, pfam PF01712) with the family of deoxyribonucleoside kinases. These enzymes catalyze the phosphorylation of deoxyribonucleoside substrates using nucleotide triphosphates as phosphate donors[29,30]. However, no kinase activity has been described until now for either NDUFA10 or complex I. Recently, Agip et al. identified a clear cryo-EM density for a nucleotide in the dNK domain of mouse NDUFA10, and modelled it as ADP[27]. A similar density had been earlier observed but not modelled in the human protein structure[48]. In the high-resolution ovine structure, the phosphorylated S36 residue, and an AMP molecule were modelled[32]. These models partially agree with our finding that a purine deoxyribonucleotide, dGTP, is tightly bound to NDUFA10 with a predicted 1:1 binding stoichiometry in mouse tissue. However, our UV-crosslink assays with purified NDUFA10 show specific binding for deoxyguanosine-containing nucleotides with the highest affinity for dGTP. Furthermore, ADP could not compete dGTP binding in our affinity assays with native mitochondrial extracts and dGTP-resin. As described above, both ovine and mouse cryo-EM densities are in principle also compatible with dGTP binding instead of earlier modelled ADP and AMP. Currently the structural data is not definitive in resolving the exact nature of the nucleotide. However, preferential binding of deoxyguanosine-containing nucleotides may be explained by the potentially strong H-bond between R104 side-chain and the guanine oxygen atom (indicated in Fig. 4b, d), which would be absent in AMP or ADP (Fig. 4a, c). This arginine is conserved in mammals, unlike S36, possibly indicating the importance of such interaction.

All key residues at the dNK domain of deoxyribonucleoside kinases are present in NDUFA10 and its active site would be accessible to solution. Interestingly, based on structural information from canonical deoxyribonucleoside kinases, Agip et al. determined that the nucleotide binding NDUFA10 would be oriented in a feedback-inhibitor mode, with its nucleoside-part binding the substrate-binding site, and its phosphates binding the P-loop[27]. In fact, this mode of inhibition was described for kinases in the dNK family that are commonly regulated by the end-product triphosphate of their preferred substrates[28]. This binding is strong (high affinity) and stable through protein purification procedures, and dNKs have been previously purified

from tissues tightly bound to their respective feedback inhibitors[43,49]. Considering the amount of protein-bound dGTP, the mitochondrial NDUFA10 content and a likely stoichiometry of 1:1 for the interaction, 62% of available NDUFA10 would be bound to dGTP in mouse liver (Figs. 1, 2b, and 5e). Thus, assuming NDUFA10 would behave as a canonical deoxyribonucleoside kinase, a substantial part of the enzyme would be inhibited in mouse mitochondria, and this could be a reason why no phosphorylation activity has been hitherto identified for complex I. NDUFA10 shares crucial residues for substrate-specificity with purine deoxyribonucleoside kinases[42,50]. In addition, considering dGTP as a possible end-product inhibitor, deoxyguanosine appeared as a likely substrate for NDUFA10. Hence, we performed our kinase assay after pre-exposing the immunopurified protein to an excess of deoxyguanosine, in an attempt to displace dGTP from the putative substrate-binding site. However, we were unable to detect dGuo phosphorylation by immunopurified NDUFA10 under our experimental conditions. Some kinase activity could still be below our detection limit due to suboptimal assay conditions, lack of essential cofactors, etc. Likewise, we cannot exclude the possibility that NDUFA10 would need to be assembled into a fully or partially constituted complex I to function.

On the other hand, salvage of purine and pyrimidine deoxyribonucleosides is already guaranteed in mitochondria by the action of other members of the dNK family: deoxyguanosine kinase and thymidine kinase 2. Alternatively, dGTP could be contributing to NDUFA10 or even complex I stability. However, our data on the NDUFA10[E160A/R161A] mutant, which shows lower dGTP binding capacity without altering NADH dehydrogenase activity or complex I assembly, suggests a different role for the protein-nucleotide interaction, at least in human cells. In agreement with this idea, structural information on the bovine and, putatively ovine, NDUFA10 revealed a phosphorylation in Ser-36 that may interfere with dGTP binding while being still compatible with complex I function. This residue is substituted by other amino acids in the human and murine protein[27,51,52]. Nonetheless, a fine-tuning of complex I activation state or assembly by dGTP binding could still be possible in different metabolic conditions.

While ribonucleotides are common regulators of biological processes, such as signal transduction through protein interaction, dNTPs are rarely reported in roles other than as DNA precursors[53–56]. dNTPs are thought to be mostly soluble in the mitochondrial matrix where they should be available for polymerases and other enzymes acting at mtDNA replication and repair processes. Interaction with these enzymes is supposed to occur transiently. In addition, free dNTPs are highly unstable and easily degraded during long experimental procedures, likely due to the activity of cellular kinases, phosphatases, and other interfering enzymes. Nonetheless, we have shown that in mitochondria from differentiated mouse tissues (brain and liver), dGTP is mostly bound to NDUFA10, and this interaction is stable throughout mitochondria isolation and protein solubilization and methanol precipitation. It is reasonable to assume that only free dNTPs are available as substrates for DNA polymerases and thus have an impact on DNA synthesis. However, if proven dynamic, and because it affects such an important proportion of mitochondrial dGTP, the NDUFA10-dGTP interaction may represent a mechanism regulating nucleotide availability. Mutation of the putative dGTP binding site in human cells (NDUFA10[E160A/R161A]) led to a 50% decrease in total mitochondrial dGTP and thus altered dNTP relative amounts in mitochondria, but this drop had no detectable effect on mtDNA copy number nor induced the generation of multiple deletions. More subtle defects, such as increased point mtDNA mutations or ribonucleotide

misincorporation, were not explored in this study and are still possible. However, our studies with the NDUFA10[E160A/R161A] were performed in highly replicative transformed cells (293T-HEK) where de novo synthesis is supposed to efficiently nourish nuclear and mitochondrial DNA synthesis. We anticipate that disrupting dGTP binding to NDUFA10 would have a greater impact on quiescent cells or differentiated tissues, where mitochondria depend mainly on slower deoxyribonucleoside salvage to replenish their dNTP pools. This option will have to be investigated in future studies.

Some other pieces of evidence suggest a link between complex I and mtDNA maintenance. MtDNA is packaged in DNA-protein structures denoted nucleoids. These structures are believed to guarantee that all proteins and elements necessary for DNA expression and maintenance are constitutively or transiently located in the vicinity of the mtDNA molecule[57–59]. Nucleoids have been repeatedly reported in association with mitochondrial membrane[59], and an assembly factor of complex I, NDUFS6, has recently been found within the nucleoid proteome[60]. Integration of complex I and other OXPHOS components into the nucleoid is considered a mechanism to help meet the high-energy demands of mtDNA transcription, replication, and translation processes. However, we propose NDUFA10 could have evolved from the pre-existing family of deoxyribonucleoside kinases (dGK, dCK, and TK2-like) to act as a dGTP reservoir in mitochondria, protecting dGTP from degradation and conferring a mechanism for rapidly expanding mitochondrial dGTP pool on the cell. Even an impact on the cytosolic dNTP pool could be possible after dGTP release in mitochondria through its direct transport across the mitochondrial membrane[10,21,22]. Further research is needed to determine whether dGTP association to NDUFA10 is modulated in vivo in response to physiological stimuli or metabolic stress, and if this can significantly contribute to the free nucleotide pool and thus DNA synthesis. If so, the NDUFA10-dGTP interaction should be viewed as the first direct link between oxidative metabolism and DNA synthesis.

## Methods

**Cell lines and culture**. HEK-293T-NDUFA10[KO] (KO) were generated by CRISPR-mediated gene editing on HEK-293T cells as described in Stroud et al[15]. HEK-293T-NDUFA10[E160A/R161A] (KI) and HEK-293T-NDUFA10[FLAG] cells were generated by introducing the flag-tagged modified NDUFA10 cDNAs in HEK-293T-NDUFA10[KO][15]. NDUFA10 WT, KO, and KI cells were grown in high-glucose (4.5 g/L) Dulbecco's Modified Eagle Medium (DMEM) with GlutaMAX™ (Thermo Fisher Scientific), supplemented with 110 mg/mL sodium pyruvate, 10% fetal bovine serum (FBS), 100 U/mL penicillin-streptomycin, and 50 µg/mL uridine.

HeLa-NDUFA10[FLAG] cells were generated by lentiviral transduction. To obtain lentiviral vectors for NDUFA10[FLAG], we co-transfected a T75 flask ($20 \times 10^6$ cells) of HEK-293T cells with lentiviral transfer vector (pLKO.1-Puro-CMV-NDUFA10[FLAG] or pLKO.1-Puro-CMV), together with pMISSIONgagpol (packaging plasmid expressing the viral gag, pol, tat, and rev genes, (packaging vector #1, Sigma-Aldrich)) and pMISSIONvsvg (envelope plasmid expressing VSV-G glycoprotein (packaging vector #2, Sigma-Aldrich)) using 1 µg/cm² PEI 25 K transfection reagent (Polysciences Inc. #23966). The transfer vectors, pMISSIONgagpol, and pMISSIONvsvg were combined at a ratio of 15:4:1 respectively. Cell culture supernatants were harvested 72 h, 80 h, and 88 h post-transfection, pooled, passed through a 0.45-µm pore size filter to remove cell debris, and stored at 4 °C. Processed supernatants were used to transduce HeLa cells. A total of $18 \times 10^6$ cells were seeded 24 h prior to transduction. 48 h after transduction, we started selection of transformed HeLa cells through puromycin treatment (1 µg/mL) for one to two weeks. After several passages, expression of Flag-tagged NDUFA10 was monitored by western blot, and integration of pLKO.1-Puro-CMV-NDUFA10[FLAG] and HeLa pLKO.1-Puro-CMV vectors was verified by PCR amplification of NDUFA10[FLAG] cDNA and puromycin resistance cassette.

Unmodified HEK-293T, HeLa and HeLa-NDUFA10[FLAG] cells were grown in high-glucose (4.5 g/L) DMEM supplemented with 10% fetal bovine serum (FBS), 2 mM L-glutamine, and 100 U/mL penicillin-streptomycin.

All cells were incubated at 37 °C and a humidified atmosphere of 5% CO₂.

HEK-293T cells were originally purchased from the ATCC and modified in Dr. Ryan's lab as described in Stroud et al.[15]. HeLa cells were obtained from

collaborators and transformed with lentivirus for NDUFA10-FLAG expression and purification.

**Mice housing and experimentation**. All procedures with animals in this study were carried out in accordance with the regulations established by the Generalitat de Catalunya for the Care and Use of Laboratory Animals. Mice were housed and bred in a standard controlled environment under a 12-h light-dark cycle with ad libitum access to water and regular rodent chow diet. The protocol was approved by the Ethics Committee for Animal Experimentation of the Vall d'Hebron Research Institute (Permit Number: 73/19) and by the Generalitat de Catalunya. 8- to 12-week-old C57BL/6 male mice were used in this study.

**Mitochondria isolation from mouse tissues and mammalian cells**. We obtained mitochondrial fractions from fresh mouse tissues by differential centrifugation. Briefly, 8- to 12-week-old mice were sacrificed by cervical dislocation, and tissues (liver, and brain) were rapidly dissected and chilled in homogenization buffer A for liver (320 mM sucrose; 1 mM ethylenediaminetetraacetic acid [EDTA]; 10 mM Tris-HCl pH 7.4) and homogenization buffer AT for brain (75 mM sucrose; 225 mM D-sorbitol; 1 mM ethylene glycol-bis (β-aminoethyl ether)-N,N,N′, N′-tetraacetic acid [EGTA]; 0.1% bovine serum albumin-free fatty acid-free [BSA FFA-free]; 10 mM Tris-HCl pH 7.4). All further operations were carried out at 2–4 °C. Organs were cut in small pieces, homogenized in a Dounce homogenizer, and centrifuged at $1,000 \times g$ for 10 min at 4 °C. The resulting supernatants were transferred to 50-mL centrifuge tubes. The pellets were either discarded (liver) or resuspended in buffer AT for a second homogenization and centrifugation round (brain). Supernatants were collected and centrifuged at $9,000 \times g$ for 10 min at 4 °C. The mitochondrial pellets were washed in 8 or 4 volumes of homogenization buffer A and AT, respectively. Finally, we resuspended the mitochondrial-enriched pellets in incubation buffer (25 mM sucrose; 75 mM sorbitol; 100 mM KCl; 10 mM K₂HPO₄; 0.05 mM EDTA; 5 mM MgCl₂; 10 mM Tris-HCl pH 7.4) for protein determination and further processing.

To obtain mitochondria from mammalian cells, we used ten 150 mm diameter plates at confluence. We obtained cell pellets by trypsinization and centrifugation at $450 \times g$ for 5 min at 4 °C. All further operations were carried out at 2–4 °C. We washed the pellets twice with ice-cold phosphate-buffered saline (PBS) and resuspended them in homogenization buffer C (250 mM sucrose; 1 mM EGTA; 10 mM Hepes pH 7.4). After homogenization with a Dounce homogenizer, we centrifuged at $1,500 \times g$ for 10 min at 4 °C. The supernatant was transferred to a 15-mL centrifuge tube, and the pellet was homogenized and centrifuged a second time. Both supernatants were combined in a 50-mL centrifuge tube and centrifuged at $10,000 \times g$ for 10 min at 4 °C. The resulting mitochondrial-enriched pellet was washed twice in homogenization buffer C, once in incubation buffer, and finally resuspended in incubation buffer for protein determination.

Protein concentration in mitochondrial fractions from either mouse tissue or cultured cells was determined by Bradford assay (Pierce™ Coomassie Plus Assay Kit, Thermo Fisher Scientific).

**Whole-cell protein extraction**. Cell pellets ($1.5 \times 10^6$ cells), obtained as previously described, were resuspended and homogenized in RIPA buffer (1% IGEPAL CA-630; 0.1% SDS; 50 mM Tris-HCl; 150 mM NaCl; 0.5% sodium deoxycholate; 1 mM EDTA) for 30 min on ice. Cell lysates were centrifuged at $20,000 \times g$ 10 min at 4 °C to eliminate cell debris and protein concentration was determined by Bradford assay.

**SDS-PAGE and western blot analysis**. Proteins from whole cell or mitochondrial lysates (10–50 µg) were resolved by electrophoresis in 12% polyacrylamide gels and wet-electrotransferred to PVDF membranes. Proteins of interests were analyzed by immunodetection with primary antibodies (see Supplementary Table 2 for a complete list of antibodies and used working dilutions): Anti-NDUFA10 antibody (GeneTex # GTX114572); Anti-Core II Complex III (Molecular Probes #A11143); Anti-COX IV (Abcam #ab16056); Anti-SDHA70 (Abcam #ab14715); Anti-VDAC (Abcam #ab15895); Anti-TFAM (GeneTex #GTX103231); Anti-39kDa subunit Complex I (Molecular Probes #A21344); Anti-FLAG M2 antibody (Sigma-Aldrich #F3165); Anti-GAPDH (Origene #TA802519). Primary antibodies were detected with secondary antibodies conjugated with horseradish peroxidase (HRP) (polyclonal rabbit anti mouse Ig HRP (Dako #P0280) and polyclonal goat anti-rabbit Ig HRP (Dako #P0448) and Immobilon Western Chemiluminescent HRP Substrate (ECL) (Millipore #WBKLS0500). Images were captured by Odyssey FC System (LI-COR Biosciences), and protein levels were quantified with Image Studio Lite Ver 5.2 software.

**Blue native polyacrylamide gel electrophoresis (BN-PAGE)**. Mitochondrial pellets (30–50 µg) were lysed for 15 min on ice at 2 µg/µL in 1% DDM, 1.75 M aminocaproic acid, 75 mM BisTris (pH 7.0), and 2 mM EDTA. The lysate was then centrifuged at $20,000 \times g$ for 20 min at 4 °C. After that, Serva Blue G (SBG) solution (750 mM aminocaproic acid; 5% Coomassie Brilliant Blue G-250) was added up to a final concentration of 0.25% G-250 to the supernatants and protein complexes were resolved by electrophoresis in a 4–16% gradient Native PAGE Bis-Tris Gels (Thermo Fisher Scientific)[61]. Electrophoresis was performed at 40 V for 15 min at

4 °C and continued at 100 V until complete resolution of respiratory chain protein complexes (approximately 4–5 h). Proteins were transferred to PVDF membranes by overnight wet-transfer and decorated with the antibodies of interest.

**dGTP-affinity purification and pull-down assays**. We used fresh isolated mitochondria from either cells or mice (10- to 25-week-old C57Bl6/J males) as previously indicated. Mitochondrial pellets were lysed at 1 mg/mL in binding buffer B (75 mM Hepes pH 7.9; 200 mM NaCl; 5% glycerol; 1 mM dithiothreitol [DTT]; 2% n-dodecyl β-D-maltoside [DDM]; 1× cOmplete™ protease inhibitor cocktail without EDTA [Roche]; 1× phosphatase inhibitor cocktail PhosSTOP™ [Roche]) for 15 min on ice. The lysate was centrifuged at $20,000 \times g$ for 10 min at 4 °C and further precleared by incubation with blank agarose (Jena Bioscience #AC-001L) 45 min at 4 °C and 15 min at room temperature. The precleared supernatants (INPUT) were incubated with blank agarose (Jena Bioscience #AC-001L), immobilized γ-amino-octyl-dGTP (Jena Bioscience #AC-112L), or γ-amino-octyl-dCTP (Jena Bioscience #AC-108L) for 15 min at room temperature followed by 45 min at 4 °C for affinity purification. Resins were pre-equilibrated by washing in at least 3 volumes of binding buffer prior to incubation with lysates. All steps involving bead transfer were performed by centrifugation at $10,000 \times g$ for 2 min at 4 °C. After binding, the flow-through (OUTPUT) was preserved and agarose beads were washed for 5 min in washing buffer (75 mM Hepes pH 7.9; 200 mM NaCl; 5% glycerol; 1 mM DTT; 0.05% DDM; 1× cOmplete™ protease inhibitor cocktail without EDTA [Roche]; 1× phosphatase inhibitor cocktail PhosSTOP™ [Roche]) at 4 °C as follows: $4 \times 5$ mL, $1 \times 3$ mL and $1 \times 1$ mL. Later, we treated beads for 30 min at room temperature with at least two-bead volumes of elution buffer (75 mM Hepes pH 7.9; 85 mM dGTP unless otherwise indicated; 200 mM NaCl; 1 mM DTT; 1× cOmplete™ protease inhibitor cocktail without EDTA [Roche]; 1× phosphatase inhibitor cocktail PhosSTOP™ [Roche]) to ensure buffer exchange and allow complete elution of bound proteins. The final ELUATES were obtained by centrifugation of beads at $1,000 \times g$ for 2 min at 4 °C.

Protein from eluates was precipitated after overnight incubation with 15% TCA and 0.2% sodium deoxycholate. The following day, the samples were centrifuged at $20,000 \times g$, washed with ice-cold acetone, air-dried and resuspended in 2× SDS loading buffer (125 mM Tris pH 6.8; 4% SDS; 20% glycerol; 200 mM DTT; 1% bromophenol blue). pH was adjusted with 1 M Tris pH 9. Proteins were further analyzed by sodium dodecyl sulphate polyacrylamide gel electrophoresis (SDS-PAGE) and subsequent gel staining or western blot analysis.

For large-scale affinity purification assays, we performed the pull-down on 5 mg lysates using 60 μL/mg of each agarose. Reduced-scale pull-down assays were performed at a 30 μL/mg ratio of agarose beads. See Supplementary Table 3 for a complete list of immunoprecipitation reagents and tools used in the study.

**Protein identification by liquid chromatography-electrospray ionization-tandem mass spectrometry (LC-ESI-MS/MS)**. Pre-visualization of affinity-purified proteins was performed by SDS-PAGE and silver staining (PlusOne Silver staining kit [GE Healthcare]) of small-scale purifications. For identification of pulled-down proteins, large-scale affinity-purified proteins were resolved by SDS-PAGE and visualized by Coomassie Blue G250 staining. Briefly, resolved gels were incubated for 1 h in fixing solution (45% methanol; 1% acetic acid) and stained with Coomassie Blue G250 solution (17% (NH₄)₂SO₄; 0.1% Coomassie Blue G250; 34% methanol; 0.5% acetic acid) overnight. Excessive staining was removed by further washing of gels with distilled water the next day. After staining, four majority bands were visualized and excised from the gel (Bands 1–4, Supplementary Fig. 2). Proteins in the corresponding gel slices were digested using modified porcine trypsin (Promega). Stained gel fragments were cut into small pieces, washed with 200 μL of 50 mM ammonium bicarbonate/50% ethanol 200 μL for 20 min and dehydrated with 200 μL of ethanol for 20 min. Reduction and alkylation were performed by incubating samples with 200 μL of 10 mM DTT in 50 mM ammonium bicarbonate for one hour at 56 °C, followed by alkylation with 200 μL of 55 mM iodoacetamide in 50 mM ammonium bicarbonate for 30 min, protected from light. The gel pieces where then washed with 200 μL of 25 mM ammonium bicarbonate for 20 min, and dehydrated with 100 μL of acetonitrile for 10 min. The acrylamide pieces were rehydrated and fully covered in a 2.7 ng/μL modified porcine trypsin (Promega) solution in 25 mM ammonium bicarbonate solution (40 μL). After overnight trypsin digestion at 37 °C, peptides were extracted by incubation with 20 μL of acetonitrile for 15 min at 37 °C, followed by 30 min at room temperature with 130 μL of 0.2% trifluoroacetic acid (TFA). The eluted peptides were SpeedVac-dried and stored at −20 °C until LC–MS/MS analysis.

Tryptic digests from excised bands were analyzed using a linear ion trap Velos-Orbitrap mass spectrometer (Thermo Fisher Scientific). Instrument control was performed using Xcalibur software package, version 2.2.0 (Thermo Fisher Scientific). Peptide mixtures were fractionated by on-line nanoflow liquid chromatography using an EASY-nLC system (Proxeon Biosystems, Thermo Fisher Scientific) with a two-linear-column system. Digests were loaded onto a trapping guard column (EASY-column, 2 cm long, ID 100 μm and packed with Reprosil C18, 5 μm particle size from Proxeon, Thermo Fisher Scientific) at 4 μL/min. Then, samples were eluted from the analytical column (EASY-column, 10 cm long, ID 75 μm and packed with Reprosil, 3 μm particle size from Proxeon, Thermo Fisher Scientific). Separation was achieved with a mobile phase of 0.1% formic acid in water (Buffer A') and acetonitrile with 0.1% formic acid (Buffer B'), and applying a

linear gradient from 0 to 35% of buffer B' for 120 min at a flow rate of 300 nL/min. Ions were generated by applying a voltage of 1.9 kV to a stainless steel nano-bore emitter (Proxeon, Thermo), connected to the end of the analytical column, on a Proxeon nano-spray flex ion source.

The LTQ Orbitrap Velos mass spectrometer was operated in data-dependent mode. A scan cycle was initiated with a full-scan MS spectrum (from m/z 300 to 1600) acquired in the Orbitrap with a resolution of 30,000. The 20 most abundant ions were selected for collision-induced dissociation fragmentation in the linear ion trap when their intensity exceeded a minimum threshold of 1000 counts, excluding singly charged ions. Accumulation of ions for both MS and MS/MS scans was performed in the linear ion trap, and the AGC target values were set to $1 \times 10^6$ ions for survey MS and 5000 ions for MS/MS experiments. The maximum ion accumulation time was 500 and 200 ms in the MS and MS/MS modes, respectively. The normalized collision energy was set to 35%, and one microscan was acquired per spectrum. Ions subjected to MS/MS with a relative mass window of 10 ppm were excluded from further sequencing for 20 s. For all precursor masses a window of 20 ppm and isolation width of 2 Da was defined. Orbitrap measurements were performed enabling the lock mass option (m/z 445.1200) for survey scans to improve mass accuracy.

LC-MS/MS data were analyzed using the Proteome Discoverer software (Thermo Fisher Scientific) to generate mgf files. Processed runs were loaded to ProteinScape software (Bruker Daltonics) and peptides were identified using Mascot (Matrix Science, London UK) to search the SwissProt database, restricting taxonomy to mouse proteins. MS/MS spectra were searched with a precursor mass tolerance of 10 ppm, fragment tolerance of 0.8 Da, trypsin specificity with a maximum of 2 missed cleavages, cysteine carbamidomethylation set as fixed modification and methionine oxidation as variable modification. Significance threshold for the identifications was set to $p < 0.05$ for the probability-based Mascot score, minimum ions score of 20, and the identification results were filtered to 1% false discovery rate (FDR) at peptide level, based on searches against a Decoy database.

**Complex I immunoprecipitation**. Mitochondrial pellets (5 mg) from mouse liver were lysed at 5 mg/mL in DDM buffer for 30 min on ice. The lysate was centrifuged at $25,000 \times g$ for 30 min at 4 °C (INPUT). The resulting supernatant was then divided and incubated in an orbital shaker overnight at 4 °C with pre-equilibrated Complex I Immunocapture Kit (Abcam #ab109711) beads (150 μg antibody/mg) or the equivalent amount of Immobilized Protein G Agarose beads (Abcam #ab174816) as a blank control. Next day, we separated FLOW-THROUGH aliquots from both incubations and washed the beads $1 \times 8$ mL and $1 \times 1$ mL with washing buffer (PBS; 0.05 % DDM; 1× cOmplete™ protease inhibitor cocktail without EDTA [Roche]; 1x phosphatase inhibitor cocktail PhosSTOP™ [Roche]. Finally, a 1/15 fraction of beads (ELUATES) was separated for western-blot monitoring of pulled-down proteins, and the remaining beads were processed for TCA-dNTP extraction.

**ANTI-FLAG immunoprecipitation**. Mitochondrial pellets (15 mg) from HeLa-NDUFA10^FLAG and HEK293T-NDUFA10^E160A/R161A cells were lysed at 1 mg/mL in lysis buffer B (50 mM Tris-HCl pH 7.4; 150 mM NaCl; 1% Triton™ X-100; 0.5% sodium deoxycholate) supplemented with 1× cOmplete™ protease inhibitor cocktail without EDTA (Roche), and 1× phosphatase inhibitor cocktail PhosSTOP™ (Roche). The lysates were precleared with 20 μL/mg of Mouse IgG Agarose (Sigma-Aldrich # A0919) for 1 h at 4 °C in an orbital shaker to reduce non-specific binding. After centrifugation at $1,000 \times g$ for 5 min at 4 °C, the beads were discarded and the precleared lysate was incubated with ANTI-FLAG M2 Magnetic Beads (30 μL/mg) (Sigma-Aldrich # M8823) for 2 h at 4 °C in an orbital shaker. ANTI-FLAG M2 Magnetic Beads were washed $5 \times 10$ mL and $1 \times 1$ mL with lysis buffer B supplemented with 1× cOmplete™ protease inhibitor cocktail without EDTA (Roche). Finally, protein was eluted with 1 beads-volume of a 3xFLAG peptide at 150 ng/μL in TBS buffer (50 mM Tris-HCl pH 7.4; 150 mM NaCl) supplemented with 1x cOmplete™ protease inhibitor cocktail without EDTA (Roche). Eluted proteins and beads were stored at −80 °C until further experiments, and immunoprecipitation (IP) efficiency was monitored by western blot. See Supplementary Table 3 for a complete list of immunoprecipitation reagents and tools used in the study.

**Fractionation by sucrose density gradient ultracentrifugation**. A total of 5 mg of mitochondrial pellets were lysed at 5 mg/mL for 30 min on ice in 1% DDM buffer (phosphate buffered saline (PBS) pH 7.4; 1% DDM; 1× cOmplete™ protease inhibitor cocktail without EDTA [Roche]; 1× phosphatase inhibitor cocktail PhosSTOP™ [Roche] or Triton™ X-100 buffer (50 mM Tris-HCl pH 7.4; 150 mM NaCl; 1% Triton™ X-100; 0.5% sodium deoxycholate; 1× cOmplete™ protease inhibitor cocktail without EDTA [Roche]; and 1× phosphatase inhibitor cocktail PhosSTOP™ [Roche]). Later, the lysates were centrifuged at $20,000 \times g$ for 30 min at 4 °C. The resulting supernatants were subsequently layered onto a 10 mL 15–37.5% discontinuous sucrose gradient made up in Tris-HCl buffer (50 mM Tris-HCl pH 7.4 with 0.05% DDM or 0.05% Triton™ X-100 depending on lysate type) and centrifuged at 38,000 rpm for 16 h at 4 °C in a Sorvall TH-641 rotor (Thermo Fisher Scientific). A total of 1-mL fractions were then collected from the bottom of the gradient. All fractions were then processed for TCA-dNTP extraction. A 20-μL

aliquot was separated from each individual fraction for subsequent western-blot monitoring.

**dNTP extraction**. For total cellular dNTP extraction, cells were grown in 150-mm dishes. When confluence was reached, we washed each dish twice with ice-cold PBS and scraped the cells with a rubber policeman and 1 mL of PBS. All operations were conducted at 2–4 °C. Cell pellets were obtained by centrifugation at $450 \times g$ for 5 min at 4 °C. The pellets were washed in 1 mL of PBS, and an aliquot was separated for protein concentration determination. Cells were again centrifuged, and an acid dNTP extraction was performed. Briefly, the pellet was treated with 600 μL of 0.5 M trichloroacetic acid (TCA) and centrifuged at $20,000 \times g$ for 5 min at 4 °C. The supernatants were neutralized with 1.5 volumes of 0.5 M trioctylamine in Freon (1,1, 2-trichlorotrifluoroethane) and centrifuged at $10,000 \times g$ for 10 min at 4 °C. A total of 300 μL of the upper aqueous phase (half the total aqueous phase) were collected to prevent contamination from the organic phase. The neutralized aqueous extracts were dried by speed vacuum and stored at –80 °C until further analysis.

Pellets corresponding to 0.5–1 mg of mitochondrial protein were processed by different methods for comparison: (1) By resuspension in 300 μL of 0.5 M TCA or 0.6 M perchloric acid (PCA), followed by neutralization with 1.5 volumes of 0.5 M trioctylamine in Freon. Half the neutralized aqueous phase was later dried by speed vacuum and stored at –80 °C until further analysis (TCA and PCA methods). (2) By resuspension in 300 μL of 60% ice-cold methanol and storage at –20 °C for 2 h followed by boiling for 5 min and centrifugation at $20,000 \times g$ for 10 min at 4 °C. Supernatants were dried by speed vacuum and stored at –80 °C until further analysis (MeOH + B + C). (3) By resuspension in 300 μL of 60% ice-cold methanol and storage at –20 °C for 2 h followed by centrifugation at $20,000 \times g$ for 10 min at 4 °C and then boiling of supernatants for 5 min. Precipitated material was pelleted by a second centrifugation at $20,000 \times g$ for 10 min at 4 °C, and final supernatants were dried by speed vacuum and stored at –80 °C until further analysis (MeOH + C + B).

To separate the protein bound and free nucleotide fractions, we used a modified MeOH + C + B method. Briefly, mitochondrial pellets (0.5–1 mg of protein) were treated with 300 μL of ice-cold 60% methanol and kept at −20 °C. After 2 h, the protein was pelleted at $20,000 \times g$ for 10 min at 4 °C. The resulting pellets (protein-bound fraction) were resuspended again in 300 μL of 60% methanol, and then treated together with the supernatants (free nucleotide fraction) with 60 μL of 3 M TCA up to a final concentration of 0.5 M. Both fractions were later centrifuged at $20,000 \times g$ for 5 min at 4 °C. Supernatants were neutralized with 1.5 volumes of 0.5 M trioctylamine in Freon and centrifuged at $10,000 \times g$ for 10 min at 4 °C. Half the aqueous phase was collected and dried by speed vacuum and stored at –80 °C until further analysis.

**Quantification of dNTPs**. For dNTP quantification, we reconstituted speed vacuum-dried mitochondrial or whole cell extracts in 40 mM Tris HCl pH 7.4. We used a radiometric polymerase-based method as in González-Vioque et al[24,62] with slight modifications. Briefly, we performed a polymerase extension reaction with Thermosequenase enzyme (GE Healthcare). We used a short common primer oligonucleotide annealed to a longer-hanging template oligonucleotide (Supplementary Table 4). The template sequence is designed so that each specific dNTP from the sample will incorporate into newly synthesized DNA, along with tritiated dATP or dTTP added to the reaction mix (Supplementary Table 5). The primer extension reaction was run at 48 °C for 1 h. Then, an aliquot of the reaction was spotted on diethylaminoethyl (DEAE) filtermat filters (PerkinElmer), dried, and washed 3 × 10 min and 3 × 5 min with 5% sodium phosphate monobasic mono-hydrate, 1 × 10 min with distilled water and 1 × 10 min with 70% ethanol. After the filters were air-dried, they were immersed in 2 mL of Ultima Gold (PerkinElmer) scintillation liquid, and incorporated radioactivity was detected in a TriCarb (PerkinElmer) scintillation counter. The dNTP concentration in the samples was extrapolated from the radioactive signal obtained from known concentration dNTP standards determined in parallel.

**Photoaffinity labeling**. Purified NDUFA10[FLAG] or NDUFA10[E160A/R161A] (35 ng in both cases) were incubated in 2× Binding Buffer A (150 mM Tris-HCl pH 7.4; 10 mM MgCl₂; 0.8 mM DTT) containing 20 nM of radiolabeled dNTP ([α-³²P]-dNTP) (EasyTide, PerkinElmer). Cold competitors were added at a concentration of 2 μM when indicated. The mixture was UV-crosslinked for 30 min at 4 °C (Hoefer UVC 500 UV crosslinker). Then, protein was precipitated overnight by adding 0.02% of sodium deoxycholate and 15% of TCA. The following day, the mixture was centrifuged at $20,000 \times g$ for 30 min at 4 °C, and the protein pellet was washed with ice-cold acetone. The protein pellets were resuspended in 20 μL of 2× SDS buffer and resolved by SDS-PAGE. After electrophoresis slab gels were pre-soaked for 5 min in fixing solution (3% glycerol; 25% ethanol) and air-dried overnight between cellophane sheets on GelAir Drying Frames (BIO-RAD #1651775). Results were obtained by autoradiography.

**Electrophoresis mobility assay (EMSA)**. Mouse liver mitochondria were lysed for 15 min at 4 °C in 100 mM phosphate buffer (KH₂PO₄), 1% DDM, 1× cOmplete™ protease inhibitor cocktail without EDTA (Roche), and 1× phosphatase inhibitor cocktail PhosSTOP™ (Roche). After centrifugation at $20,000 \times g$ for 20 min at 4 °C, 10 μg of the mitochondrial extract were incubated with 0.06 pmol of [α-³²P]-dGTP (3000 Ci/mmol) in binding buffer (75 mM Hepes pH 7.9; 50% glycerol; 180 mM NaCl; 2 mM DTT; 2 mM phenylmethylsulfonyl fluoride, and 2 μg/μL BSA) for 30 min at 30 °C. Cold competitors were added at 100-molar excess 30 min prior to [α-³²P]-dGTP addition. Samples were run in native 6% acrylamide:bisacrylamide (37.5:1) gels. After electrophoresis, gels were vacuum-dried on a filter paper and developed by autoradiography.

**MtDNA analysis**. mtDNA deletions were investigated by long-range PCR and mtDNA copy number was assessed by quantitative PCR. Total DNA was isolated from cultures of NDUFA10[WT], NDUFA10[KO] and NDUFA10[KI] cells using the QIAamp DNA Mini Kit (QIAGEN). DNA was dissolved in 10 mM Tris-HCl pH 8.0 and quantified by spectrophotometry (NanoDrop Spectrometer, Thermo Scientific). Long-range PCR was performed using the LA Taq polymerase (Takara) as described by Nigishaki et al.[63]. Real-time PCR was performed using an ABI PRISM 7900HT real-time system (Applied Biosystems). Detection of mtDNA and nuclear DNA was performed as a multiplex PCR reaction with TaqMan Universal PCR Master Mix II with UNG (Applied Biosystems), and custom-designed sets of TaqMan probes and primers (12 S rRNA gene for mtDNA, and RNase P single copy gene for nuclear DNA). Analyses were performed using the SDS 2.4 software (Applied Biosystems)[18]. Primers and probes used for mtDNA copy number determination are listed in Supplementary Table 6.

**Respiratory chain complex I activity**. A total of $1 \times 10^6$ cells were resuspended in 200 μL of mannitol buffer (10 mM Tris-HCl pH 7.2; 225 mM mannitol; 75 mM sucrose; 0.1 mM EDTA) and homogenized by sonication for 5 s at 60% intensity with Microson Ultrasonic Cell Disruptor XL2000 (Misonix). Then, the homogenates were centrifuged at $650 \times g$ for 20 min at 4 °C, and the protein concentration of the resulting supernatants was quantified by Bradford assay. Homogenates were brought to 1 mg/ml in mannitol buffer. NADH oxidation was determined after electron transfer to ubiquinone by monitoring the decrease in 340 nm absorbance[64,65]. Briefly, 20 μg of protein homogenate were mixed with 100 μM decilubiquinone, 3.75 mg/mL BSA, and 50 mM K₂HPO₄ pH 7.5 in the presence or absence of 12.5 μM of rotenone. The reaction was initiated by adding 100 μM NADH at 37 °C, and light absorbance was monitored at 340 nm for 3 min in a UV-2401PC UV-VIS recording spectrophotometer (Shimadzu Corporation).

**Oxygen consumption determination**. Oxygen consumption and extracellular acidification rates were measured in live cells using a Seahorse Bioscience XFe-96 Analyzer. Briefly, 25,000 cells were plated per well in culture plates treated with 50 μg/mL poly-D-lysine. For each assay cycle, there were 4 measurements of 2 min mix, 2 min wait and 5 min measure. The following inhibitor concentrations were used: 2 μM oligomycin, 0.5 μM carbonyl cyanide m-chlorophenyl hydrazone (CCCP), 0.5 μM rotenone and 0.3 μM antimycin A. Data were normalized using the CyQUANT assay kit (Invitrogen) and analyzed using the Wave2.6.0 and Prism (v8.0.2, GraphPad) software packages.

**Deoxyguanosine kinase activity determination**. NDUFA10[FLAG] was FLAG-affinity purified from HeLa-NDUFA10[FLAG] as above indicated but including a washing step with 1 mM deoxyguanosine (dGuo) prior to final elution from ANTI-FLAG M2 Magnetic Beads. We adapted Park and Ives's et al.[28,66] methodology for determination of dGuo kinase activity. In summary, 15 ng of immunoprecipitated protein in elution buffer (50 mM Tris pH 7.6; 0.5% Triton X-100; 150 mM NaCl; 20% glycerol; 100 ng/μl 3XFLAG peptide) were incubated with 1 volume of 2x reaction buffer (100 mM Tris pH 7.6; 20 mM ATP; 20 mM MgCl₂; 20 mM DTT; 1% Triton X-100; 40 μM (2'-deoxyguanosine, [8-³H]; [Vitrax]) in a final volume of 40 μL for 5 min at 37 °C. Following the addition of MgCl₂ up to 5 mM, the samples were incubated for 1 h at 37 °C. After the incubation, 30 μL of the reaction mix were spotted on Whatman DE81 filters (GE Healthcare) and air-dried. Subsequently, filters were washed three times for 5 min in 5 mM ammonium formate and placed in vials to elute with 2 mL of 0.1 M HCl, 0.2 M NaCl for 30 min at room temperature. 14 mL of UltimaGOLD Scintillant Liquid (Amersham Biosciences) was added to the eluate, and radioactivity was measured in a TRICARB scintillation counter (Perkin Elmer).

In parallel, protein extracts were obtained from mouse liver and used as a control for deoxyguanosine kinase activity detection. Briefly, a microspoon of liquid nitrogen-pulverized tissue was homogenized in extraction buffer (50 mM Tris pH 7.6; 2 mM DTT; 1% Triton X-100; 2 mM EDTA; 1x cOmplete™ protease inhibitor cocktail without EDTA [Roche]), using a microtube homogenizer. After incubation on ice for 20 min, this was centrifuged at $20,000 \times g$ for 20 min at 4 °C, and the supernatant was recovered. The protein concentration was determined by Bradford assay, and 25 μg of protein were mixed in elution buffer. To prevent deoxyguanosine degradation by catabolic enzymes in the tissue sample, 1 μM Immucillin H (inhibitor of purine nucleoside phosphorylase; gift from Prof. Vern Schramm, Albert Einstein College of Medicine, New York, USA) was added to the reaction mix; 50 mM deoxycytidine was also added to competitively inhibit [³H]-dGuo phosphorylation by deoxycytidine kinase (dCK) present in the tissue.

**Structural analysis**. The listed PDB files and corresponding cryo-EM maps were downloaded from PDB and EMDB databases. The PDB and CIF parameter files for dGTP were generated in GRADE web server using the PDB entry DGT as a starting point. Previously placed ADP or AMP ligands were replaced by the dGTP, which was real-space refined into cryo-EM density using COOT. Resulting structures and maps were visualised using PyMOL.

**Statistics and reproducibility**. Statistical analysis was performed with the GraphPad Prism 6 software. The specific tests and number of independent experiments used for each assay are indicated in the Results section or the Figure legends. For statistical purposes, undetectable values were considered zero.

**Multiple alignment**. Protein sequences were obtained from the Uniprot database (hTK2 (O00142-1); hdCK (P27707-1); hdGK (Q16854-1); hNDUFA10 (O95299-1); DmdNK (Q9XZT6-1); DrTK2 (A0A0R4IP35-1); DrdGK (A5WUY2)). Multiple alignment of protein sequences was performed using T-coffee software (http://tcoffee.crg.cat/) followed by edition with the Boxshade tool (http://www.ch.embnet.org/software/BOX_form.html).

**Reporting summary**. Further information on research design is available in the Nature Research Reporting Summary linked to this article.

## Data availability

All raw data underlying the graphs and charts presented in the main and supplementary figures are present in Supplementary Data files 1 and 2, respectively. Uncropped images for western-blots are included in Supplementary Fig. 7. Radiochemicals, antibodies and other materials used in the study are listed in Supplementary Tables 2–7. Structural data used for modeling NDUFA10 binding to dGTP are available at https://www.rcsb.org (10.2210/pdb6ZKA/pdb; 10.2210/pdb6g2j/pdb). The mass spectrometry datasets generated during the dGTP pull-down studies have been deposited to the ProteomeXchange Consortium via the PRIDE[67] partner repository with the dataset identifier PXD033900. All other data are available from the corresponding authors on reasonable request.

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

## Acknowledgements

We thank Dr, Luke Formosa (Department of Biochemistry and Molecular Biology, Monash Biomedicine Discovery Institute, Monash University, Melbourne, Australia) for his valuable advice and assistance on NDUFA10 molecular studies and Dr. Francesc Canals and his team (Proteomics Laboratory, Vall d'Hebron Institute of Oncology [VHIO], Universitat Autònoma de Barcelona, Barcelona, Spain) for their assistance with LC-MS/MS analyses. This work was supported by the Spanish Ministry of Industry, Economy and Competitiveness [grants BFU2014-52618-R, SAF2017-87506, and PID2020-112929RB-I00 to Y.C.], by the Spanish Instituto de Salud Carlos III [grants PI21/00554 and PMP15/00025 to R.M.], co-financed by the European Regional Development Fund (ERDF), and by an NHMRC Project grant to M.R. (GNT1164459).

## Author contributions

D.M.G., E.G.V., M.R., Y.C., and R.M. designed the study. D.M.G., E.G.V., M.D., R.C.P., A.V.G., J.T.T., L.S., and Y.C. performed experiments and analyzed the data. D.M.G., Y.C., and R.M. wrote the manuscript. D.M.G., E.G.V., M.D., R.C.P., A.V.G., J.T.T., L.S., M.R., R.M., and Y.C. participated in the critical reading of the manuscript. Y.C. and R.M. supervised the project.

## Competing interests

The authors declare the following competing interests: R.M. and Y.C. report grants and non-financial support from Modis Therapeutics, personal fees and other from Modis Therapeutics, outside the submitted work. RM and YC have a patent "Treatment of mitochondrial diseases" (PCT/ EP2016/062636) with royalties paid by Modis Therapeutics. RM has a patent "Deoxynucleoside therapy for diseases caused by unbalanced nucleotide pools including mitochondrial DNA depletion syndromes" (PCT/US16/038110) with royalties paid by Modis Therapeutics. These relationships are de minimus for Vall d'Hebron Research Institute and CIBERER. The other authors report no conflict of interests.

## Additional information

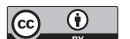

