## [Peer Review file · Communications Biology]

Reviewers' comments:

Reviewer #1 (Remarks to the Author):

Others have shown the dNTP pools in mammalian mitochondria to be asymmetric, with dGTP to be highly overrepresented. This paper explores the basis for the asymmetry and reports that most of the dGTP in mammalian mitochondria is bound to NDUFA10, an accessory subunit of complex I of the respiratory chain. Mutagenesis of the deoxynucleoside kinase domain of NDUFA10 yields a protein with diminished dGTP binding in vitro and diminished dGTP pool in cultured cells. This result strongly suggests that most of the apparent excess dGTP in mammalian mitochondria is bound to the NDUFA10 protein. The authors speculate that this finding suggests a link between oxidative metabolism and mitochondrial DNA maintenance.

This is an outstanding paper. The authors have established a basis for a puzzling finding, with the results suggesting an important relationship between DNA metabolism and energy generation within mitochondria. The data are clearly presented, and the manuscript is refreshingly free of minor errors. The few errors I detected are not sufficient, in my opinion, to delay publication of the paper.

Reviewer #2 (Remarks to the Author):

Comments :

- Some more precisions could be given in the abstracts, as for example the mutant used and some values on decreased dGTP content in cells expressing the mutant.
- In introduction: I would not say that de novo synthesis is based on RNR, it is rather dependent on this enzyme.
- Page 3: The two first sentences in the last paragraph should have references, unless this is from the current study and then this should be indicated.
- I believe (Fig. EV1) should be (Fig S1). Same for other supplemental figures.
- The second part of figure S2 is a table and should be presented as one.

We thank the positive comments from both reviewers that have considered our work was of interest for its publication in *Communications Biology*. We have addressed the few minor concerns raised only by referee #2 (all introduced changes are highlighted in green in the edited manuscript text). Please see below for a point-by-point response to the referees.

Ramon Martí
Yolanda Cámara

Reviewer #1 (Remarks to the Author):

Others have shown the dNTP pools in mammalian mitochondria to be asymmetric, with dGTP to be highly overrepresented. This paper explores the basis for the asymmetry and reports that most of the dGTP in mammalian mitochondria is bound to NDUFA10, an accessory subunit of complex I of the respiratory chain. Mutagenesis of the deoxynucleoside kinase domain of NDUFA10 yields a protein with diminished dGTP binding in vitro and diminished dGTP pool in cultured cells. This result strongly suggests that most of the apparent excess dGTP in mammalian mitochondria is bound to the NDUFA10 protein. The authors speculate that this finding suggests a link between oxidative metabolism and mitochondrial DNA maintenance.

This is an outstanding paper. The authors have established a basis for a puzzling finding, with the results suggesting an important relationship between DNA metabolism and energy generation within mitochondria. The data are clearly presented, and the manuscript is refreshingly free of minor errors. The few errors I detected are not sufficient, in my opinion, to delay publication of the paper.

Reviewer #2 (Remarks to the Author):

Comments :

- Some more precisions could be given in the abstracts, as for example the mutant used and some values on decreased dGTP content in cells expressing the mutant.

We shortly exceed the upper word limit required by the editorial, so we have limited our edits to the strictly suggested by the reviewer. Consequently, we now mention the particular mutations introduced in NDUFA10, and the percent of dGTP reduction observed in the mutant cell's mitochondria.

- In introduction: I would not say that de novo synthesis is based on RNR, it is rather dependent on this enzyme.

We agree with the referee so we have substituted "*is based on*" by "*depends on*" in the introduction section to be more accurate with the description.

- Page 3: The two first sentences in the last paragraph should have references, unless this is from the current study and then this should be indicated.

We have introduced an explanatory sentence, '*In the present study, we have found*' to make clear the mentioned results are described in the present study and can not be found in previous literature.

- I believe (Fig. EV1) should be (Fig S1). Same for other supplemental figures.

We have corrected the mistaken nomenclature for all supplementary figures that are now referred as 'Supplementary Figure X'.

- The second part of figure S2 is a table and should be presented as one.

We have split prior Fig S2 into 'Supplementary Figure 2' and 'Supplementary Table 1' as suggested.